# PASO: STEP PARALLEL STOCHASTIC OPTIMIZATION

## ABSTRACT

This paper approaches the fundamental challenge of accelerating the inherently autoregressive nature of gradient descent (GD) like SGD and Adam through a dynamic system perspective. Specifically, we introduce a unified framework that recasts the autoregressive GD process as solving a system of triangular nonlinear equations (TNEs), thereby facilitating a paradigm shift toward non-autoregressive GD featuring parallel gradient computation across iteration steps. Within this generic framework, we establish that: (1) the TNE system admits a unique solution corresponding precisely to the autoregressive GD iterative trajectory; (2) solving the TNEs system guarantees convergence to the GD iterative trajectory in equal or far fewer iterations. Building on these insights, we present *PASO*, a step parallel optimizer for accelerating a broad class of autoregressive GD optimizers like SGD and Adam. Extensive experiments (*e.g.*, Llama-3.2-1B) validate that PASO achieves up to $91\times$ reduction in GD steps and $7.5\times$ speedup in wall-clock time, with no measurable model quality loss. We anonymously open-source our code here: https://anonymous.4open.science/r/PASO-8ECE/.

## 1 INTRODUCTION

Stochastic gradient descent (SGD) (Robbins and Monro, 1951) and its variants (Kingma and Ba, 2014; Duchi et al., 2011; Tieleman, 2012; Loshchilov and Hutter, 2017; Liu et al., 2025; Nesterov, 1983; Pagliardini et al., 2024; Hwang, 2024), continue to be fundamental optimization engines for training deep neural networks. These methods underpin breakthroughs across domains (Vaswani et al., 2017; He et al., 2016; Shi et al., 2022; Lu et al., 2024; 2023), exemplified by large language models (LLMs) (Brown et al., 2020; OpenAI, 2023a; Touvron et al., 2023) and computer vision systems (Radford et al., 2021; Rombach et al., 2022; Lu et al., 2025). At its essence, SGD operates through iterative parameter adjustments where each iterative step follows the negative gradient direction of a randomly sampled data batch. To produce high-quality models, however, SGD typically necessitates an enormous number of iterative steps involving repeated forward and backward passes through massive datasets, resulting in prolonged training durations. For instance, training modern LLMs like DeepSeek-V3 (Liu et al., 2024) and GPT-4 (OpenAI, 2023b) often demands hundreds of thousands to millions of iteration steps. As a result, training a large-scale model consumes millions of GPU hours, roughly amounting to 1-3 months or longer (OpenAI, 2023b; Liu et al., 2024). Therefore, the staggering SGD steps caused by the exponential growth of model and dataset sizes have made a fundamental efficiency bottleneck.

Existing efforts to accelerate SGD training follow two main paradigms. The first develops more advanced optimizers(Zhang et al., 2025; Robert et al., 2025; Cheng and Glasgow, 2025; Hwang, 2024; Duchi et al., 2011; Liu et al., 2025; Polyak, 1964) (e.g., Adam (Kingma and Ba, 2014)) to accelerate convergence through refined update rules. However, the reduction in iterative steps is modest at best, as these techniques continue to rely on the sequential step-by-step execution of SGD. The second strategy develops parallel SGD, where workers update the model synchronously or asynchronously. Synchronous methods, typical in distributed learning, require waiting for all nodes to compute gradients before each update, while asynchronous approaches (e.g., DC-ASGD (Zheng et al., 2017) and HOGWILD! (Recht et al., 2011a) ) allow workers to update global parameters independently. However, they risk harming model performance because of stale gradients. More critically, their parallelization is confined to intra-step operations, while leaving the inter-step sequential dependency intact, thereby maintaining the total number of training iterations unchanged.

This paper investigates a critical question: *can we drastically reduce gradient descent steps by parallelizing the step execution without sacrificing model performance?* At first glance, the challenge seems insurmountable—GD is inherently sequential, bound by a rigid Markov dependency chain (Zinkevich et al., 2010a). Surprisingly, we demonstrate that it is possible to completely sever this dependency chain to enable fully parallel gradient computation across different GD steps. In particular, we approach this problem through the lens of non-linear equations, treating the points on the GD iteration trajectory as mutually independent unknown variables. This independence thus naturally eradicates the sequential dependencies between iterations. By solving this system of equations, we achieve fully parallel gradient computation across all steps. Empirical results show this approach converges in far fewer iterations compared to standard sequential GD. Furthermore, our framework is inherently orthogonal to existing approaches, allowing seamless integration with both sequential optimizers (e.g., Adam) and parallel GD variants (e.g., model, pipeline, and data parallelism).

In summary, this paper contributes the following theoretical and practical advancements:

- we present PASO, an innovative step-level parallel GD paradigm, through transforming the autoregressive GD process into solving a system of triangular nonlinear equations;
- we establish that PASO exhibits guaranteed convergence to the GD trajectory points with iteration complexity equal to or surpassing that of the autoregressive gradient descent;
- our comprehensive evaluation showcases that PASO reduces iteration steps by up to $91\times$ and accelerates wall-clock time by $7.5\times$, all without sacrificing quality.

## 2 RELATED WORK

**Model Parallelism.** Model parallelism (Jia et al., 2019; Narayanan et al., 2021; Xu et al., 2021; Yuan et al., 2021; Rajbhandari et al., 2020; Ren et al., 2021; Xu et al., 2020; Gao et al., 2025) shards a neural network's parameters across multiple devices to accommodate models too large for a single device's memory. During training, all devices perform partial computations in parallel over its designated subset of parameters. Subsequently, through collective communication methods like NCCL (NVIDIA, 2023), the results from each device are aggregated to compute the global gradient, which is then used to update the parameters. Early works by (Dean et al., 2012; Chilimbi et al., 2014; Xing et al., 2015) introduced model parallelism by partitioning model parameters across machines. Recent advances include Mesh-TensorFlow (Shazeer et al., 2018), PyTorch's fully sharded data parallel (Zhao et al., 2023), and Megatron-LM (Shoeybi et al., 2019) which efficiently parallelizes large models to achieve substantial speedups. Despite enabling the training of enormous models, these frameworks only address model parallelization within one step of gradient descent.

**Pipeline Parallelism.** Pipeline parallelism(He et al., 2021; Kim et al., 2020; Li et al., 2021; Sun et al., 2025; Zhao et al.; Tang et al.) aims to reduce idle time by partitioning models among workers per the direction of data flow into multi-stages and processing micro-batches in an interleaved manner. GPipe (Huang et al., 2019) introduced gradient accumulation for consistency across pipeline stages, while PipeDream (Narayanan et al., 2019) improved efficiency with "1F1B" scheduling and weight stashing. Despite these innovations, pipeline parallelism maintains sequential dependencies between gradient steps, as each micro-batch must wait for previous steps' gradients to update parameters.

**Data Parallelism**. Data parallelism partitions the training data across multiple workers (e.g., GPUs or nodes), and each worker computes gradients on its local data subset. The gradients are then aggregated to update the model parameters through two primary mechanisms: synchronous SGD (SSGD) and asynchronous SGD (ASSGD). In SSGD (Zinkevich et al., 2010b; Dekel et al., 2012; 2010; Ye et al., 2022; McMahan et al., 2017), all workers compute gradients in parallel, but the parameter server waits for all workers to finish before applying the aggregated gradients to the model. This ensures consistency but may suffer from stragglers. ASSGD (Baudet, 1978; Bertsekas and Tsitsiklis, 2015; Cohen et al., 2021; Recht et al., 2011b; Feyzmahdavian and Johansson, 2023; Stich et al., 2021; Nguyen et al., 2022; Even et al., 2024; Recht et al., 2011a; Zhang et al., 2015) address this limitation by enabling independent parameter updates without synchronization. A prominent example is HOGWILD! (Recht et al., 2011a), which implements lock-free updates to the shared model parameters in memory. However, ASSGD faces challenges with gradient staleness (Dutta et al., 2018).

Current methods primarily focus on parallelization within individual GD steps, which retains the inherent limitations of traditional autoregressive GD. In contrast, PASO disrupts this autoregressive dependency chain, introducing step parallelism where multiple different GD steps are executed simultaneously. We humbly believe that PASO's step-level parallelization naturally complements existing intra-step parallelization methods, establishing a new avenue for parallel GD. We observe that recent work (Shu et al., 2024) leaves kernelized gradient estimation to enable approximately parallelized iterations; our method is compatible with it, as the kernelized gradient estimation can be used to give more precise initial points for our PASO.

## 3 PRELIMINARY

### 3.1 STOCHASTIC GRADIENT DESCENT (SGD)

Given a mini-batch $\zeta$ of size $B$, the loss function for the batch is defined as the average loss over the samples in $\zeta$:

$$\mathcal{L}(w, \zeta) = \frac{1}{B} \sum_{x,y \in \zeta} \ell(w; x, y), \tag{1}$$

where $\ell(w; x, y)$ denotes the loss for a single sample $(x, y)$. The model parameters $w$ are updated iteratively using the gradient of the batch loss:

$$w_t = w_{t-1} - \eta_{t-1} \nabla_{w_{t-1}} \mathcal{L}(w_{t-1}, \zeta_{t-1}), \tag{2}$$

where $\eta_t$ is the learning rate at iteration $t$, $\zeta_t$ stands for the mini-batch used at iteration $t$, and $\nabla_w \mathcal{L}(w_t, \zeta_t)$ is the gradient of the batch loss. Other popular optimizers, such as Adaptive Moment Estimation (Adam), also update parameters iteratively. For brevity, a detailed description of these methods is provided in Appendix G.

## 4 PROPOSED METHOD

### 4.1 MOTIVATION

Gradient descent (GD) algorithms like SGD and Adam use historical weights to compute the current weight, which is the essence of an autoregressive process. We formally define this process below.

**Definition 1** (Autoregressive GD Procedure). *Initiating with a model weight $w_0$, the GD process like SGD and Adam represents an autoregressive procedure in the specific form of*

$$w_t = w_0 - \sum_{\tau=0}^{t-1} \eta_\tau \, g_\tau \big( w_\tau, \ldots, w_{\tau-r+1}; \zeta_\tau, \ldots, \zeta_{\tau-r+1} \big), t \in \{1, \cdots, T\}, \tag{3}$$

where $1 \leq r \leq \tau + 1$ and the general gradient term $g_\tau$ is determined by the specific optimizer. For example, the $g_\tau$ for SGD depends only on the most recent weight and mini-batch (i.e., $r = 1$):

$$g_{t-1}(w_{t-1}; \zeta_{t-1}) = \nabla_{w_{t-1}} \mathcal{L}(w_{t-1}, \zeta_{t-1}). \tag{4}$$

The explicit $g_{t-1}$ formulations for more complex optimizers like Adam are detailed in Appendix G.

We observe that when all the model weights $w_0, \cdots, w_T$ are considered as unknown variables, the autoregressive GD procedure above transforms into a system of $T + 1$ nonlinear equations (NEs). By providing an initial set of guesses for the true weights, this system of NEs can be solved in parallel since there are no dependencies among the $T + 1$ NEs. As a result, the model weights $w_0, \cdots, w_T$ can be computed concurrently.

### 4.2 RECASTING AUTOREGRESSIVE GD AS TRIANGULAR NONLINEAR EQUATION SOLVING

Inspired by current parallel algorithms (Song et al., 2021; Tang et al., 2024; Lu et al., 2025), such a series of cascaded functions in Definition 1 can be regarded as a system of $T + 1$ NEs with a triangular structure. Denote by $\hat{w}_0, \cdots, \hat{w}_T$ the unknown variables corresponding to the iterative tracjectory $w_0, \cdots, w_T$ generated from the autoregressive GD process in Definition 1.

**Definition 2** (Triangular NEs). *We define the system of triangular NEs for the autoregressive procedure in Definition 1 as*

$$\mathcal{F}(\hat{w}_0, \cdots, \hat{w}_T) = \begin{cases} \hat{w}_0 - w_0 = 0, \\ \hat{w}_t - F_{t-1}(\hat{w}_0, \cdots, \hat{w}_{t-1}; \zeta_0, \ldots, \zeta_{t-1}) = 0, t \in \{1, \cdots, T\}, \end{cases} \quad (5)$$

where $F_{t-1}$ is defined as:

$$F_{t-1}(\hat{w}_0, \cdots, \hat{w}_{t-1}; \zeta_0, \ldots, \zeta_{t-1}) = \hat{w}_0 - \sum_{\tau=0}^{t-1} \eta_\tau g_\tau(\hat{w}_\tau, \ldots, \hat{w}_{\tau-r+1}; \zeta_\tau, \ldots, \zeta_{\tau-r+1}), \quad (6)$$

where $g_\tau$ depends on the choice of the specific GD algorithms. In Appendix G, we include the explicit form of $g_\tau$ for various GD algorithms like AdamW.

This formulation offers several advantages. First, it decouples the dependencies among $w_t$, enabling synchronous calculation for all gradients $\nabla_{w_t} \mathcal{L}(w_t, \zeta_t), t \in \{0, \cdots, T-1\}$. This parallelism makes the approach especially suitable for modern parallel computing infrastructures, such as distributed systems. Second, the triangular NEs have been extensively studied in mathematics, providing access to a variety of well-established methods for solving such systems efficiently.

While we can now calculate the gradients across steps in parallel by solving the triangular NEs, an important question remains: do the solutions found via equation solving yield model weights comparable to those generated by the autoregressive GD process? Specifically, can we assert that $\hat{w}_t = w_t$ holds for all $t \in [0, T]$?

**Proposition 1** (Unbiased Estimation (see *App. H* for proof)). *The TNEs system in Eq. (5) possesses a unique solution that unbiasedly estimates the GD trajectory $\{w_\tau\}_{\tau=0}^T$ in Definition 1.*

This finding demonstrates that by solving for the TNEs, a model of comparable quality to that derived from the traditional autoregressive GD process can be obtained.

### 4.3 SOLVING THE SYSTEM OF TNES

The field of optimization provides various methods for solving a system of NEs. Since our primary goal is to study a fundamental step-parallel GD optimizer, we implement only the classical fixed-point iteration (FPI) method (Banach, 1922) and postpone more advanced alternatives for future exploration. Applying FPI to find the solution of an equation system involves reformulating the equation system into an iterative form. It is easy to know the iterative form of Eq. (5) corresponds to a system with $T$ iterative components in Eq. (7). Therefore, given an initial set of guesses $\hat{w}_0^{(0)}, \cdots, \hat{w}_T^{(0)}$, and randomly sampled $T$ mini-batches $\zeta_0, \ldots, \zeta_{T-1}$, the system of FPI for the TNEs is as follows:

$$\hat{w}_t^{(k)} = F_{t-1}(\hat{w}_0^{(k-1)}, \cdots, \hat{w}_{t-1}^{(k-1)}; \zeta_0, \ldots, \zeta_{t-1}), t \in \{0, \cdots, T-1\}, \quad (7)$$

where $\hat{w}_0^{(k)} = w_0, \forall k \in \{0, \cdots, K\}$; $K$ is the number of parallel iterations. For a more intuitive FPI system, see Definition 3 in Appendix.

**Proposition 2** (Convergence Analysis (see *App. I* for proof)). *From any initial guess $\{\hat{w}_t^{(0)}\}_{t=0}^T$, the fixed-point iteration in Eq. (7) converges exactly to the autoregressive GD trajectory $\{w_t\}_{t=0}^T$ defined in Definition 1. This convergence is achieved in at most $T$ steps.*

In practice, the number of parallel iterations $K$ required for convergence is significantly smaller than $T$, resulting in substantial empirical speedups. Besides, we also provide empirical validation that the PASO trajectory is functionally equivalent to the original, with the near-zero average L2 norm and variance between them confirming a high-fidelity reproduction (*App. D.2.1*).

### 4.4 COMPUTATION-EFFICIENT SUBEQUATIONS SOLVING

Solving the above triangular NEs necessitates computing $T$ gradients $\{\nabla_{\hat{w}_t} \mathcal{L}(\hat{w}_t, \zeta_t)\}_{t=0}^{T-1}$ in parallel across the entire time horizon. For large values of $T$, this becomes computationally prohibitive when restricted to a limited number of computing nodes. To tackle this, our core idea is to perform the fixed-point iteration only on $p \leq T$ subequations per iteration via a sliding window technique in

Figure 1: Illustration of Step-parallel Training Paradigm PASO. During iteration $k$, PASO performs simultaneous weight updates across steps within a $p$-size sliding window through parallel gradient computations. The process consists of: (1) computing update terms $g^{(k)}$ based on current weights $\hat{w}^{(k)}$ after calculating their graidients in parallel; followed by (2) determining new weights $\hat{w}^{(k+1)}$.

(Shih et al., 2024). Specifically, we perform parallel equation solving only on a subset of $T+1$ NEs, within a sliding window of size $p$:

$$\hat{w}_{t+i}^{(k)} = F_{t-1+i}(\hat{w}_0^{(k-1)}, \cdots, \hat{w}_{t-1+i}^{(k-1)}; \zeta_0, \ldots, \zeta_{t-1+i}), i \in \{0, \cdots, \min\{p-1, T-1\}\}, \quad (8)$$

This window size can be tuned to match the number of available computing nodes. Additionally, the window slides forward dynamically, with the sliding distance determined by the number of equations for which solutions have been found in the current window.

## 4.5 STOPPING CRITERION

To ensure that the parallel optimizer achieves performance on par with that of the autoregressive GD process, it's essential to establish a suitable stopping criterion to assess whether the solution values are no longer changing each parallel iteration. Let $\delta$ represent the convergence tolerance threshold, governing the allowable variation in solution values between successive iterations. In accordance with (Zhou et al., 2024), we define the stopping criterion as:

$$d(\hat{w}_t^{(k)}, \hat{w}_t^{(k-1)}) := \frac{1}{n} \left\| \hat{w}_t^{(k)} - \hat{w}_t^{(k-1)} \right\|^2 \leq \delta, \quad (9)$$

where $\|\cdot\|$ denotes the Frobenius norm; $n$ is model dimension. $\delta$ is updated adaptively via exponential moving average (EMA):

$$\delta = \lambda\delta + (1 - \lambda) \cdot \mathcal{A}\Big(\big\{d(\hat{w}_{t+i}^{(k)}, \hat{w}_{t+i}^{(k-1)}) | i = 1, \cdots, p\big\}\Big), \quad (10)$$

where $\lambda$ is the EMA decay rate; $\mathcal{A}$ is a mean or median function.

## 4.6 INITIALIZATION

The parallel iteration in Eq. (8) begins with a set of initial weights $\{\hat{w}_t^{(0)}\}_{t=0}^p$. We initialize all the model parameters within the $p$-sized sliding window using the default initial weight $w_0$: $\hat{w}_t^{(0)} = w_0, \forall t \in \{0, \cdots, p\}$. When the sliding window moves forward, we initialize all the newly introduced model parameters using the rightmost weight from the previous window. We emphasize that initialization also stands as a crucial component for improving parallel efficiency. By starting iterations with weights that are close approximations of the target solutions $\{w_t\}_{t=0}^p$, more rapid convergence can be achieved. We defer this critical aspect to future research.

---

**Algorithm 1:** PASO: Step Parallel Stochastic Optimization within A Sliding Window

---

**Input** : Default model initial weight $w_0$, gradient descent steps $T$, learning rate $\{\eta_t\}_{t=0}^{T-1}$, random mini-batches $\{\zeta_t\}_{t=0}^{T-1}$, tolerance $\delta$, window size $p$, model dimension $n$, EMA decay rate $\lambda$.

**Output** : $\hat{w}_T^K$.

1 Obtain update rule $g_t(\hat{w}_t, \cdots, \hat{w}_{t-r+1}; \zeta_t, \cdots, \zeta_{t-r+1})$ by a GD algorithm.     // E.g., Eq. (4) or Eq. (15)

2 Initialize $\{\hat{w}_t^{(0)} = w_0, t = 0, \cdots, p\}$     // Initialize $p$ model weights within the sliding window.

3 $t, k \leftarrow 0, 0; k \in [0, K]$

4 **while** $t < T$ **do**

5    $\nabla_{\hat{w}_{t+i}^{(k)}} \mathcal{L}(\hat{w}_{t+i}^{(k)}, \zeta_{t+i}), \forall i \in \{0, \cdots, p-1\}$     // Compute each gradient concurrently.

6    $g_{t+i}^{(k)}, \forall i \in \{0, \cdots, p-1\}$     // Calculate updates in parallel (e.g., via Eq. (4)).

7    $w_{t+i+1}^{(k+1)} \leftarrow \hat{w}_t^{(k)} - \sum_{j=t}^{t+i} \eta_j g_j^{(k)}, \forall i \in \{0, \cdots, p-1\}$     // Update weights at iteration $k$ via Eq. (7).

8    $s \leftarrow \min\left(\{i+1; \hat{w}_{t+i+1}^{(k+1)} \text{ unsatisfying Eq. (9)}, \forall i \in \{0, \cdots, p-1\}\} \cup \{p\}\right)$     // The sliding stride.

9    $\hat{w}_{t+p+j}^{(k+1)} \leftarrow \hat{w}_{t+p}^{(k)}, \forall j \in \{1, \cdots, s\}$     // Initialize new model weights.

10    $\delta \leftarrow$ Eq. (10)     // Update tolerance via exponential moving average.

11    $t \leftarrow t + s, \quad k \leftarrow k + 1, \quad p \leftarrow \min(p, T - t)$

**Return:** $\hat{w}_T^{(K)}$

---

### 4.7 COMPLETE PASO ALGORITHM

Algorithm 1 details the complete process of the proposed PASO over a sliding window. After obtaining the autoregressive GD update rule (Line 1) and preparing an array of initial weights $\{\hat{w}_t^{(0)}\}_{t=0}^{p-1}$ via the default model weight (Line 2), PASO initiates the parallel optimization loop at Line 3 in which a batch of weights $\{\hat{w}_t^{(k)}\}_{t=0}^{p-1}$ within a sliding window undergo synchronous updating. Line 5 compute the gradients, which are the basic computational units of parallelism. The updates are computed in Line 6, and Line 7 updates the current model weights, preparing for the next parallel iteration $k + 1$. Line 8 checks the variation between new weights and the current weights and then determines the stride to which the window can slide forward. Line 9 initializes new model parameters outside the current window using the rightmost weight in the current window according to the sliding stride $s$. Fig. 1 shows the pipeline of PASO.

## 5 COMPUTATIONAL COST, MEMORY, AND SPEEDUP RATIO ANALYSIS

PASO introduces a novel *step-parallel* approach that is orthogonal to traditional parallelization paradigms. This naturally raises a key question: *under comparable computational and memory constraints, how does the speedup efficiency of PASO compare to that of conventional methods?* To this end, Table 1 provides a comprehensive comparison against existing methods, indicating three main conclusions:

- **Acceptable Overhead:** The total computational cost of PASO ($mT$) is comparable to that of model and pipeline parallelism ($T$) and significantly lower than data parallelism ($NT$). This minimal overhead is a worthwhile trade-off for the performance gains.

- **Superior Speedup:** The speedup ratio of PASO is $\frac{N}{m(1+\alpha N/p)}$. Since $m \approx 1$ and $p > 1$, PASO's speedup is strictly greater than the $\frac{N}{1+\alpha N}$ achieved by other methods. This indicates that PASO can be approximately up to **p** times faster than existing parallel approaches.

- **Better Scalability:** When the window size equals the number of GPUs ($p = N$), PASO's speedup ratio simplifies to $\frac{N}{m(1+\alpha)}$. As $N$ increases, the denominator in PASO's speedup formula grows much more slowly than in other methods (where it is dominated by the $\alpha N$ term), demonstrating PASO's superior scalability, especially in communication-bound scenarios.

In summary, under similar costs, PASO achieves a higher theoretical speedup and exhibits better scalability than existing parallel methods. We believe our step-parallel approach holds significant potential for fully leveraging modern parallel hardware for deep learning training.

Table 1: Comparison of computational cost, storage, and speedup ratio across parallel training methods. The analysis shows PASO's superior speedup potential and scalability. Denote by $N$ the number of GPUs, $\alpha \triangleq t_{\text{comm}}/t_{\text{comp}}$ the communication-to-computation time ratio, and $m \triangleq T/pK \approx 1$ empirically (see Fig. 9 in *Appendix*). Detailed derivations are available in *Appendix E.2*.

| Method | Computational Gradient Count | Storage per Device | Speedup Ratio ($S$) | Scalability Limit ($\lim_{N \to \infty} S$) |
|---|---|---|---|---|
| Sequential | $T$ | 1 model + 1 optimizer | $1$ | $1$ |
| Data Parallel | $NT$ | 1 model + 1 optimizer | $\frac{N}{1+\alpha N}$ | $1/\alpha$ |
| Model Parallel | $T$ | $\sim \frac{1}{N}$ model + 1 optimizer | $\frac{N}{1+\alpha N}$ | $1/\alpha$ |
| Pipeline Parallel | $T$ | $\sim \frac{1}{N}$ model + 1 optimizer | $\frac{N}{1+\alpha N}$ | $1/\alpha$ |
| **Step Parallel (PASO)** | $\mathbf{pK = mT}$ | **1 model + 1 optimizer** | $\frac{\mathbf{N}}{\mathbf{m(1+\alpha N/p)}}$ | $\mathbf{p}/(\mathbf{m}\alpha)$ |

# 6 EXPERIMENT

## 6.1 EXPERIMENT SETTINGS

**Dataset and Model.** We investigate our PASO on both an image classification task and a language modeling task. For the image task, we use the CIFAR-10 dataset (Krizhevsky et al., 2009) over a compact convolutional neural network (CNN) and Tiny-ImageNet dataset (Le and Yang, 2015) on a more harder ViT model. For the language modeling task, we train a GPT-2 model and 1B large model Llama-3.2-1B on the WikiText dataset. Further experimental details are provided in the appendix.

**Evaluation Metrics.** For the CIFAR-10 dataset, we evaluate the performance using six standard metrics: testing accuracy, testing precision, testing recall, testing F1-score, iterations, and wall-clock time. For the WikiText dataset, we evaluate the performance using four standard metrics: testing accuracy, testing perplexity, iterations, and wall-clock time.

**Hyperparameter Settings**. For the GPT2, Llama-3.2-1B, and ViT model, we use 8 NVIDIA A100 GPUs. For CNN, we use 8 NVIDIA 3090 GPUs. We employ the sweep function in Wandb (WandB, 2023) to investigate the influence of hyperparameters on the model's performance metrics, configuring a hyperparameter sweep with the following search ranges: tolerance threshold $\delta$ sampled uniformly in $[10^{-6}, 10^{-4}]$, EMA decay rate $\lambda$ sampled uniformly in $[0.8, 0.9999]$, and adaptivity scheme $\mathcal{A}$ chosen between *mean* and *median* operators. We found setting $\delta = 10^{-5}$ and $\lambda \in [0.9, 0.9999]$ easily yields comparable performance across models and datasets. Please refer to *Appendix* for the sweep results in Fig. 8.

## 6.2 EXPERIMENT RESULTS

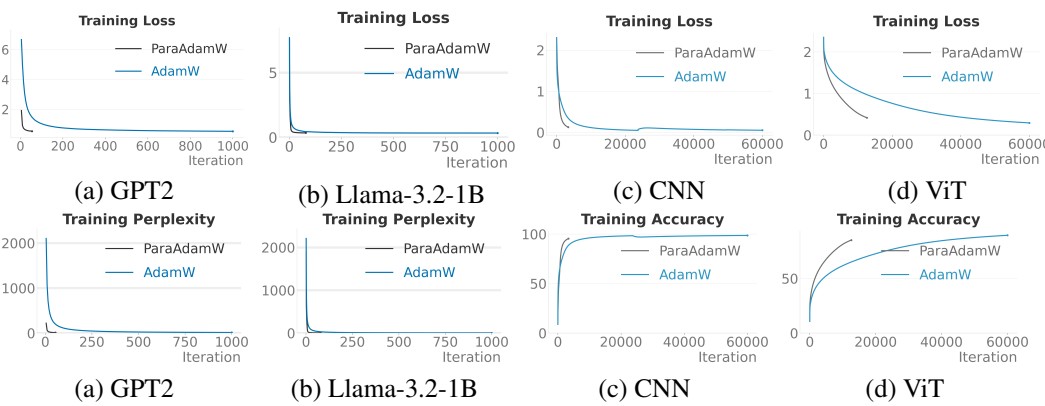

Figure 2: The comparison of loss, perplexity, and accuracy curves.

**Language Modeling Task**. Table 7 demonstrates that PASO accelerates the convergence of these optimizers without sacrificing model performance. Notably, PASO reduces the required iteration steps for sequential methods by a factor of $12.6 \sim 19.2$, resulting in a up to $7.5\times$ improvement in wall-clock time. This speedup implies that an LLM originally requiring 100 days of training can now be trained in just 13 days. Note that larger batch sizes could yield higher runtime speedups for Llama-3.2-1B, but our present implementation supports a maximum batch size of only 30.

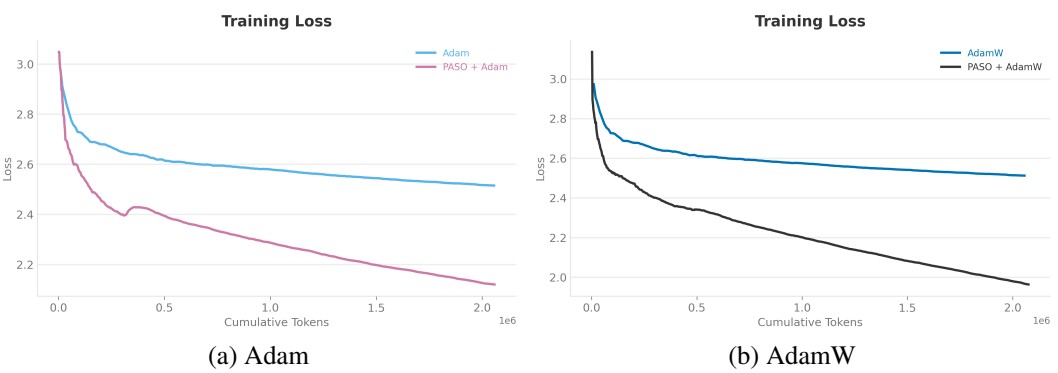

(a) Adam  (b) AdamW

Figure 3: The comparison of loss curves w.r.t. the cumulative token consumption on Llama 3.2-1B over WikiText during training. We define token count as tokens processed by the model per iteration, reporting the cumulative consumed tokens during training.

Table 2: Quantitative comparisons of different methods on WiKiText. The best results are highlighted in **bold**. "↑" (resp. "↓") means the larger (resp. smaller), the better. We define token count as tokens processed by the model per iteration, reporting the cumulative consumed tokens during training. We report the mean and standard deviation over 5 runs with different random seeds.

| Method | WiKiText, GPT-2 Model, $B = 112, \eta = 6e-5, T = 1000$ | | | | |
|---|---|---|---|---|---|
| | Iters ↓ | Total tokens ↓ | Perplexity ↓ | Time (s) ↓ | Speedup ↑ |
| SGD | 1000 | 7344101 | $36.7_{\pm 0.06}$ | 713 | 1.0× |
| SGD + PASO | **52** (19.2×) | **7339874** | $37.1_{\pm 0.11}$ | **95** | **5.5×** |
| Adam | 1000 | 7344101 | $20.4_{\pm 0.22}$ | 716 | 1.0× |
| Adam + PASO | **57** (17.5×) | **7312230** | $20.9_{\pm 0.07}$ | **106** | **5.1×** |
| AdamW | 1000 | **7344101** | $20.4_{\pm 0.06}$ | 715 | 1.0× |
| AdamW + PASO | **57** (17.5×) | 7361833 | $20.8_{\pm 0.18}$ | **107** | **5.2×** |
| Method | WiKiText, Llama-3.2-1B Model, $B = 30, \eta = 6e-5, T = 1000$ | | | | |
| | Iters ↓ | Total tokens ↓ | Perplexity ↓ | Time (s) ↓ | Speedup ↑ |
| SGD | 1000 | 2056338 | $12.6_{\pm 0.11}$ | 827 | 1.0× |
| SGD + PASO | **69** (14.5×) | **2038761** | $12.8_{\pm 0.19}$ | **266** | **3.1×** |
| Adam | 1000 | **2056338** | $12.6_{\pm 0.20}$ | 838 | 1.0× |
| Adam + PASO | **79** (12.6×) | 2058639 | $12.2_{\pm 0.05}$ | **279** | **3.0×** |
| AdamW | 1000 | **2056338** | $12.6_{\pm 0.22}$ | 854 | 1.0× |
| AdamW + PASO | **78** (12.8×) | 2061722 | $12.4_{\pm 0.09}$ | **281** | **3.0×** |

**Image Classification Task**. Table 3 compares SGD, Adam, and AdamW with their PASO-enhanced versions, showing that PASO consistently accelerates convergence while preserving model performance. For instance, in CNN model, Adam+PASO achieves a 31.2× step reduction (to 1919) with a 2.7× runtime speedup. The accuracy, precision, recall, and F1-score, confirm PASO's efficiency without compromising model quality. Note that in CV tasks, the smaller runtime speedup than the LLM tasks is due to the higher

Table 4: Impact of $p$ on perplexity.

| | Iters | Perplexity | Time (s) | Speedup |
|---|---|---|---|---|
| AdamW | 1000 | 20.6 | 1063 | 1× |
| $p = 7$ | 184 | 20.4 | 182 | 5.8× |
| $p = 35$ | 39 | 20.5 | 173 | **6.1×** |
| $p = 77$ | 21 | 20.7 | 209 | 5.1× |
| $p = 117$ | 16 | 20.8 | 235 | 4.5× |
| $p = 159$ | 12 | 20.8 | 240 | 4.4× |
| $p = 201$ | 11 | 20.9 | 267 | 4.0× |

Table 3: Quantitative comparisons of different methods on CIFAR-10. We report the mean and standard deviation over 5 runs with different random seeds.

| Method | CIFAR-10, CNN Model, $B = 4096, \eta = 1e-3$ | | | | | | |
|---|---|---|---|---|---|---|---|
| | Iters↓ | Accuracy ↑ | Precision↑ | Recall↑ | F1-score↑ | Time (s) ↓ | Speedup↑ |
| SGD | 60000 | $78.7_{\pm 0.14}$ | $78.5_{\pm 0.18}$ | $78.7_{\pm 0.12}$ | $78.6_{\pm 0.15}$ | 4277 ▰ | 1.0× |
| ParaSGD (SGD + PASO) | **1723** (34.8×) | $78.6_{\pm 0.21}$ | $78.5_{\pm 0.19}$ | $78.7_{\pm 0.16}$ | $78.5_{\pm 0.20}$ | **1339** ▰ | **3.2×** |
| Adam | 60000 | $81.3_{\pm 0.11}$ | $81.4_{\pm 0.15}$ | $81.3_{\pm 0.12}$ | $81.3_{\pm 0.13}$ | 4223 ▰ | 1.0× |
| ParaAdam (Adam + PASO) | **1919** (31.2×) | $81.7_{\pm 0.25}$ | $81.7_{\pm 0.22}$ | $81.7_{\pm 0.24}$ | $81.7_{\pm 0.23}$ | **1574** ▰ | **2.7×** |
| AdamW | 60000 | $81.5_{\pm 0.17}$ | $81.5_{\pm 0.16}$ | $81.5_{\pm 0.18}$ | $81.5_{\pm 0.17}$ | 4267 ▰ | 1.0× |
| ParaAdamW (AdamW + PASO) | **1924** (31.2×) | $81.6_{\pm 0.19}$ | $81.5_{\pm 0.21}$ | $81.6_{\pm 0.18}$ | $81.5_{\pm 0.20}$ | **1573** ▰ | **2.7×** |
| Method | CIFAR-10, ViT Model, $B = 2048, \eta = 1e-5$ | | | | | | |
| | Iters↓ | Accuracy ↑ | Precision↑ | Recall↑ | F1-score↑ | Time (s) ↓ | Speedup↑ |
| SGD | 60000 | $77.2_{\pm 0.28}$ | $77.1_{\pm 0.31}$ | $77.2_{\pm 0.29}$ | $77.0_{\pm 0.30}$ | 36305 ▰ | 1.0× |
| ParaSGD (SGD + PASO) | **3975** (15.1×) | $77.6_{\pm 0.25}$ | $77.3_{\pm 0.22}$ | $77.6_{\pm 0.24}$ | $77.3_{\pm 0.23}$ | **10481** ▰ | **3.5×** |
| Adam | 60000 | $81.6_{\pm 0.15}$ | $81.7_{\pm 0.12}$ | $81.6_{\pm 0.14}$ | $81.5_{\pm 0.13}$ | 36282 ▰ | 1.0× |
| ParaAdam (Adam + PASO) | **4219** (14.2×) | $81.3_{\pm 0.20}$ | $81.4_{\pm 0.18}$ | $81.4_{\pm 0.19}$ | $81.3_{\pm 0.21}$ | **11185** ▰ | **3.2×** |
| AdamW | 60000 | $82.0_{\pm 0.10}$ | $82.0_{\pm 0.11}$ | $82.0_{\pm 0.10}$ | $82.0_{\pm 0.10}$ | 36310 ▰ | 1.0× |
| ParaAdamW (AdamW + PASO) | **4231** (14.2×) | $81.9_{\pm 0.14}$ | $82.0_{\pm 0.13}$ | $81.9_{\pm 0.15}$ | $82.0_{\pm 0.14}$ | **11208** ▰ | **3.2×** |

communication-to-computation ratio, as CV models are smaller and compute faster per step, making communication overhead more significant.

**Impact of the Window Size** $p$. Tab. 4 illustrates the influence of the window size $p$ on the speedup of AdamW with 1000 steps over the GPT-2 model with batch size 130 and learning rate $6e-5$. As $p$ increases, the number of iterations needed for convergence significantly drops, from 184 to 11, yielding a step reduction ranging from $4.0\times$ to **91**×. This suggests that we can achieve up to $91\times$ walk-clock time acceleration without loss of model quality using 201 GPUs. However, due to our current constrained GPU resources, larger $p$ values introduce higher computational overhead per GPU, leading to increased wall-clock time. We believe that with more computing cores, the time speedup of PASO could be substantially unleashed.

**Impact of Batch Size**. As shown in Tab. 5 on GPT-2 model with learning rate $6e-5$, speedup of PASO becomes more significant as the batch size increases, while maintaining performance comparable to the baseline. This is because a larger batch size allows for better utilization of the GPU's computing capabilities for large-scale matrix operations, thereby improving parallel efficiency. More ablation studies on EMA decay rate and tolerance are shown in *Appendix* D.2.4.

**Impact of Learning Rate**. Figure 4 illustrates that PASO performs robustly across a practical range of learning rates from $4 \times 10^{-4}$ to $1 \times 10^{-2}$. These rates are comparable to those in standard optimizers

Table 5: Impact of batch size ($B$) on perplexity (PPL) and Accuracy (ACC). Top to bottom: $B = 10, 50, 90, 130$.

| Method | Iters | PPL | Time (s) | Speedup |
|---|---|---|---|---|
| AdamW | 1000 | 18.5 | 146 | 1× |
| PASO | 64 | 18.4 | 61 | 2.4× |
| AdamW | 1000 | 19.6 | 301 | 1× |
| PASO | 57 | 19.3 | 90 | 3.3× |
| AdamW | 1000 | 20.1 | 822 | 1× |
| PASO | 56 | 20.2 | 121 | 6.8× |
| AdamW | 1000 | 20.5 | 1063 | 1× |
| PASO | 56 | 20.9 | 155 | **6.9×** |

(e.g., Adam's default of $1 \times 10^{-3}$), demonstrating
that the model can converge effectively without additional limitations on learning rates.

## 7 LIMITATIONS AND FUTURE DIRECTIONS

While PASO achieves significant runtime acceleration ($2\times$–$7.5\times$), this remains substantially below its step-level speedup (up to $91\times$). Two primary factors limit performance: (1) our constrained GPU resources that inherently restrict maximum acceleration, and (2) our current implementation exists inefficient inter-GPU communication requiring model transfers to pass through a CPU intermediary. These limitations are further exacerbated by unoptimized handling of gradient synchronization, load imbalance, and kernel launch overheads, and so on.

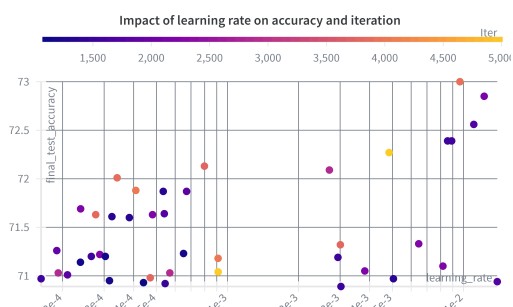

Figure 4: Impact of learning rate on accuracy and iterations. The total steps are 10000. Darker points indicate faster convergence.

Looking ahead, PASO's efficiency can be substantially improved through: (1) algorithmic refinements like gradient compression to reduce overhead; (2) system-level enhancements such as collective operations (e.g., NCCL all-reduce) to alleviate bottlenecks. These advancements could position PASO as a promising paradigm for more efficient parallel training, with broader implications for large-scale deep learning.

## 8 CONCLUSION

This paper introduces PASO, a novel framework that accelerates stochastic optimization by reformulating its autoregressive process as a system of triangular nonlinear equations (TNEs), enabling step parallel gradient computation. Theoretically, we prove that the TNE system has a unique solution matching the stochastic optimization's iteration trajectory, and solving it converges as efficiently as or faster than sequential method. Empirically, PASO achieves up to $91\times$ speedup in steps without quality degradation.

### ETHICS & REPRODUCIBILITY STATEMENTS

**Ethics Statement.** This work presents a fundamental methodology for accelerating stochastic optimization algorithms. To the best of our knowledge, it does not raise any immediate ethical concerns. The research is theoretical and empirical in nature, based on mathematical analysis and standard benchmark tasks. We do not employ any private or sensitive data. However, we acknowledge that any optimization technology has the potential for dual use. We encourage the community to utilize this work responsibly.

**Reproducibility Statement.** We are committed to fostering reproducible research. To this end:

- The theoretical claims in this paper, including the uniqueness of the solution to the triangular nonlinear equations and the convergence guarantees, are supported by formal proofs provided in appendix.

- The empirical results are obtained using standard datasets and benchmarks. To ensure reproducibility, we will open-source the complete implementation of the PASO framework, including scripts for all experiments.

- The code package will include detailed documentation, instructions for setting up the computational environment, and scripts to replicate the reported speedup and performance comparisons against sequential baselines.

- All hyperparameters and experimental settings are explicitly documented in the paper's experimental section.

We believe these measures will enable other researchers to verify our findings and build upon this work.

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

CONTENTS

## A    COMPLEXITY ANALYSIS FOR OPTEX AND PASO

To understand the trade-offs inherent in step-parallel optimization, this section provides a comparative complexity analysis of PASO and OptEx. We analyze the total computational, space, and communication costs incurred during each "main iteration" of the respective algorithms.

- For PASO, a "main iteration" refers to a single parallel iteration $k$ of its fixed-point solver (Algorithm 1).

- For OptEx, a "main iteration" refers to a single sequential iteration $t$ that dispatches $N$ parallel gradient computations.

**Definitions**. We define the following variables for our analysis:

- $d$: The dimensionality of the model's parameter vector.
- $p$: The parallelism degree (window size) of PASO.
- $N$: The parallelism degree (number of parallel steps) of OptEx.
- $T_0$: The size of the local gradient history used by OptEx.
- $G$: The number of available processing units (e.g., GPUs).
- $t_{grad}(d)$: The computational cost of a single gradient calculation (one forward and backward pass), which is typically proportional to $d$. We simplify its cost to $\mathcal{O}(d)$.
- $t_{comm}(k, d)$: The communication cost to exchange $d$-dimensional vectors among $k$ nodes (e.g., via All-Gather or All-Reduce).

For simplicity, we assume an ideal scenario where the parallelism degree matches the number of available GPUs (i.e., $G = p$ for PASO and $G = N$ for OptEx).

## A.1 OptEx Complexity Analysis

**Total Computation (per iteration $t$).** The total computational cost for OptEx is the sum of its core gradient computation and its overhead:

1. **Core Gradient Computation**: $N$ gradients are computed in parallel. With $G = N$ GPUs, this takes $\mathcal{O}(\lceil N/G \rceil \cdot t_{grad}(d)) = \mathcal{O}(d)$.
2. **Overhead Computation**: This consists of two serial steps:
    - *Kernelized Gradient Estimation*: Building the estimation model $\mu_t$ from $T_0$ history points involves $T_0 \times T_0$ kernel matrix operations (e.g., inversion), costing $\mathcal{O}(T_0^3)$, plus matrix-vector operations costing $\mathcal{O}(T_0 d)$.
    - *Multi-Step Proxy Updates*: Serially computing $N - 1$ proxy steps to find the inputs for the parallel computation. Each step requires evaluating the kernel model, costing $\mathcal{O}(T_0 d + T_0^2)$. The total cost for this stage is $\mathcal{O}(N(T_0 d + T_0^2))$.

Combining these, the total computational cost is $\mathcal{O}(d + (T_0^3 + T_0 d) + (NT_0 d + NT_0^2))$. This is dominated by the overhead, yielding $\mathcal{O}(T_0^3 + NT_0(d + T_0))$.

**Total Space (excl. base model)** OptEx must store the $T_0$ historical gradients ($\mathcal{O}(T_0 d)$), the $T_0 \times T_0$ kernel matrix ($\mathcal{O}(T_0^2)$), and the $N$ inputs for the proxy updates ($\mathcal{O}(Nd)$). The total space is $\mathcal{O}(T_0 d + T_0^2 + Nd)$.

**Total Communication (per iteration $t$)** The primary communication involves gathering the $N$ computed gradients from the $N$ GPUs, which costs $\mathcal{O}(t_{comm}(N, d))$. The proxy updates are serial and do not add to the inter-node communication.

## A.2 PASO Complexity Analysis

**Total Computation (per iteration $k$)** The total computational cost for PASO is the sum of its parallel gradient computation and its serial update overhead:

1. **Core Gradient Computation**: $p$ gradients are computed in parallel. With $G = p$ GPUs, this takes $\mathcal{O}(\lceil p/G \rceil \cdot t_{grad}(d)) = \mathcal{O}(d)$.
2. **Overhead Computation**: This consists of serial updates on the host/main node after the parallel gradients are collected:
    - *Calculate Update Term $g^{(k)}$ (Line 6)*: For stateful optimizers like Adam, this forms a serial dependency chain of length $p$, costing $\mathcal{O}(pd)$.

- *Update Model Weights $\hat{w}^{(k+1)}$ (Line 7)*: Updating all $p$ weights requires a cumulative sum, resulting in $\sum_{i=0}^{p-1}(i+1) \approx \mathcal{O}(p^2)$ vector additions. The total cost is $\mathcal{O}(p^2d)$.
- *Check Error (Line 8)*: Computing $p$ L2-norm differences costs $\mathcal{O}(pd)$.

Combining these, the total computational cost is $\mathcal{O}(d + pd + p^2d + pd) = \mathcal{O}(p^2d)$.

**Total Space (excl. base model)**    PASO needs to store the $p$ weights in the current window ($\mathcal{O}(pd)$) and their corresponding optimizer states (e.g., $p$ sets of $m, v$ moments for Adam, also $\mathcal{O}(pd)$). The total space is $\mathcal{O}(pd)$.

**Total Communication (per iteration $k$)**    The primary communication involves gathering the $p$ computed gradients from the $p$ GPUs, which costs $\mathcal{O}(t_{comm}(p, d))$. The serial updates occur locally on the main node.

### A.3    SUMMARY AND DISCUSSION

Table 6: Total per-iteration complexity comparison of OptEx and PASO, assuming parallelism degree matches the number of available GPUs ($G = p = N$). The speedup rate for OptEx is cited from the original paper, while ours is obtained from our Table 1 1).

| Metric | OptEx | PASO |
|---|---|---|
| Parallelism Degree | $N$ | $p = N$ |
| Total Computation | $\mathcal{O}(T_0^3 + NT_0(d + T_0))$ | $\mathcal{O}(N^2d)$ |
| Total Space (Overhead) | $\mathcal{O}(T_0d + T_0^2 + Nd)$ | $\mathcal{O}(Nd)$ |
| Total Communication | $\mathcal{O}(t_{comm}(N, d))$ | $\mathcal{O}(t_{comm}(N, d))$ |
| Speedup rate | $\mathcal{O}(\sqrt{N})$ | $\mathcal{O}(N)$ |

We summarize the complexity analysis in Table 6. The analysis reveals several key differences:

1. **Computational Cost**: OptEx's computation is highly sensitive to the history size $T_0$, featuring a $\mathcal{O}(T_0^3)$ term independent of model dimension $d$. In contrast, PASO's cost is independent of any history $T_0$ but scales quadratically with its parallelism degree $p$ ($\mathcal{O}(p^2d)$). For large models (where $d$ is a dominant factor), the comparison simplifies to $\mathcal{O}(NT_0d)$ for OptEx versus $\mathcal{O}(p^2d)$ for PASO.

2. **Space Cost**: PASO demonstrates a significant advantage in space complexity. Its space overhead $\mathcal{O}(pd)$ scales linearly with its parallelism, whereas OptEx requires storing both the history and the proxy inputs, resulting in a larger $\mathcal{O}((T_0 + N)d)$ footprint.

3. **Communication Cost**: The communication costs are analogous and scale with their respective parallelism degrees, $N$ and $p$.

4. **Faster Speedup**: PASO achieves a linear speedup rate of $\mathcal{O}(N)$ with respect to the number of GPUs $N$. In contrast, OptEx only achieves a sub-linear speedup of $\mathcal{O}(\sqrt{N})$. This indicates that as the number of parallel processors (GPUs) increases, PASO's efficiency scales proportionally, whereas OptEx's performance gains diminish significantly.

This analysis suggests that PASO and OptEx have similar computational and communicational complexities. However, OptEx relies on a kernel estimation method that necessitates storing $T_0$ historical gradient records, which makes its space complexity much higher than that of PASO (about $\frac{T_0+N}{N}$ times). This requirement, leading to a memory overhead proportional to $\mathcal{O}(T_0d)$, makes OptEx infeasible for large-scale models where storage (memory) is often the primary bottleneck.

## B    VALIDATION AGAINST APPROXIMATE STEP-PARALLEL METHODS OPTEX

We conduct a direct empirical comparison against a representative state-of-the-art approximate step-parallel method, **OptEx** Shu et al. (2024). Following Shu et al. (2024), we evaluate all methods on

the three synthetic benchmark functions used in their official implementation[1]: the Sphere function, the Ackley function, and the Rosenbrock function (please refer to Appendix B.2.1 in the **OptEx** paper for the details of the three functions).

## B.1 EXPERIMENTAL SETUP

We strictly adhere to the experimental setup provided in the OptEx codebase. As stated, the original OptEx source code only provides a complete and verifiable implementation for these three synthetic tasks. Due to the complexity of OptEx's kernel function estimation, we limit our comparison to these functions to ensure a correct and fair implementation of the OptEx baseline.

For all experiments, we use the Adam optimizer Kingma and Ba (2014) as the base solver. The standard, sequential Adam optimizer serves as our **Vanilla** baseline. We integrated our PASO framework directly into the *optex_cmp.py* script from the OptEx repository, ensuring that all methods (Vanilla, OptEx, and PASO) share the same experimental harness, initialization, and objective function calls.

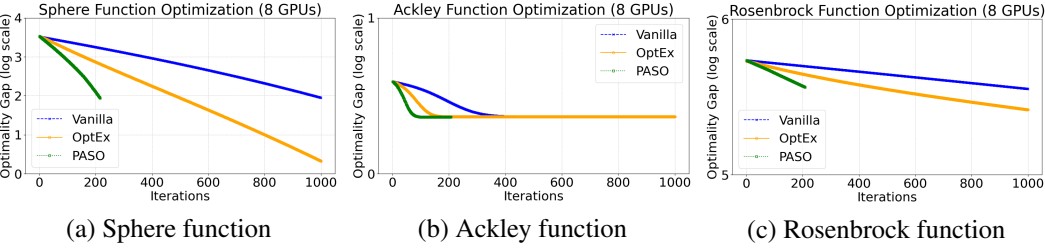

| (a) Sphere function | (b) Ackley function | (c) Rosenbrock function |

Figure 5: The comparison of optimality gap when optimizing the Sphere function, the Ackley function, and the Rosenbrock function.

## B.2 METRICS AND RESULTS

The primary metric for comparison is the **Optimality Gap**, defined as the final objective value $f(x_T)$ achieved by the Vanilla (Adam) baseline after $T = 1000$ iterations. (Note: for all three functions, the true minimum is $f^* = 0$, so the final loss $f(x_T)$ is equivalent to the optimality gap).

We then measure the number of sequential **Iterations** (i.e., communication rounds) required for OptEx and PASO to reach this same final optimality gap. The **Speedup** is calculated as the ratio of iterations taken by the Vanilla baseline (1000) to the iterations required by the parallel method.

The quantitative results are presented in Table 7 and the complete training processes are shown in Figure 5. The results clearly demonstrate the superior efficiency of PASO. The source code can be found in our anonymous code repository: https://anonymous.4open.science/r/PASO-8ECE/.

- **On the Sphere function**, all methods converge to the same optimality gap of 1.95. However, PASO requires only **214** iterations, achieving a **4.67×** speedup. This is **2.36×** faster than OptEx, which requires 504 iterations (1.98× speedup).

- **On the Ackley function**, the performance gap is even more pronounced. PASO reaches the target gap of 0.4 in just **86** iterations, resulting in a massive **11.63×** speedup. OptEx requires 162 iterations (6.17× speedup).

- **On the complex Rosenbrock function**, PASO maintains a significant advantage, achieving a **4.76×** speedup (210 iterations) compared to OptEx's 2.08× speedup (479 iterations) to reach the 5.5 optimality gap.

Across all three standard benchmarks, PASO consistently and significantly outperforms OptEx, requiring far fewer communication rounds to converge to the same solution quality. This highlights the effectiveness of PASO's adaptive error-based stopping criterion, which achieves a more efficient parallel rollout compared to OptEx's kernel-based approximation approach.

---

[1]https://github.com/youyve/OptEx

Table 7: Quantitative comparisons of different methods. The best results are highlighted in **bold**. "↑" (resp. "↓") means the larger (resp. smaller), the better. We report the mean and standard deviation of the optimality gap over 5 independent runs.

| Method | Sphere function, $d = 1e4, \eta = 1e-3, T = 1000$ | | |
| --- | --- | --- | --- |
| | Optimality gap ↓ | Iters ↓ | Speedup ↑ |
| Adam | $1.95 \pm 0.12$ | 1000 | $1.0\times$ |
| OptEx | $1.95 \pm 0.08$ | 504 | $1.98\times$ |
| PASO | $1.95 \pm 0.15$ | **214** | **4.67×** |

| Method | Ackley function, $d = 1e4, \eta = 1e-3, T = 1000$ | | |
| --- | --- | --- | --- |
| | Optimality gap ↓ | Iters ↓ | Speedup ↑ |
| Adam | $0.40 \pm 0.18$ | 1000 | $1.0\times$ |
| OptEx | $0.40 \pm 0.06$ | 162 | $6.17\times$ |
| PASO | $0.40 \pm 0.11$ | **86** | **11.63×** |

| Method | Rosenbrock function, $d = 1e4, \eta = 1e-3, T = 1000$ | | |
| --- | --- | --- | --- |
| | Optimality gap ↓ | Iters ↓ | Speedup ↑ |
| Adam | $5.50 \pm 0.09$ | 1000 | $1.0\times$ |
| OptEx | $5.50 \pm 0.17$ | 479 | $2.08\times$ |
| PASO | $5.50 \pm 0.04$ | **210** | **4.76×** |

## C    USE OF LLM

During the preparation of this work, we used Large Language Models (LLMs) to assist with the writing process. The primary uses included polishing and improving the fluency of the text, generating preliminary drafts of proofs, and assisting in the creation and formatting of tables. After using these tools, the author(s) reviewed and edited the content extensively. We take full responsibility for the entire content of this publication, including the ideas, proofs, and presentations ultimately contained in the final manuscript.

## D    MORE EXPERIMENTAL DETAILS

We evaluate PASO over two popular model training tasks, including image classification and text generation model. The results of these experiments demonstrate that PASO enhances the efficiency of autoregressive GD methods by approximately 1.5 times, all while maintaining consistent model quality as measured by metrics like accuracy or perplexity.

### D.1    EXPERIMENT SETTINGS

**Dataset and Model.** We investigate our PASO on both an image classification task and a language modeling task. For the image task, we use the CIFAR-10 dataset (Krizhevsky et al., 2009), a widely recognized benchmark in computer vision. The dataset comprises 60,000 $32 \times 32$ RGB images spanning 10 distinct classes, divided into 50,000 training and 10,000 test samples. The small image size and real-world noise make CIFAR-10 a challenging yet efficient testbed for lightweight models. We evaluate our approach by training a compact Convolutional Neural Network (CNN), following a standard architecture with convolutional and pooling layers, as shown in Table 8. Besides, we train a Vision Transformer (ViT) model [2] on CIFAR-10. For the language task, we train GPT-

---

[2]https://www.modelscope.cn/models/iic/multi-modal_clip-vit-large-patch14_336_zh

2 model[3] and Llama-3.2-1B[4] on the WikiText dataset[5], a large-scale corpus of Wikipedia articles preprocessed for language modeling. The dataset is publicly available and commonly used for training and benchmarking autoregressive models like GPT-2. We adopt the standard GPT-2 architecture, leveraging Hugging Face's `transformers` library for tokenization and training loops. We assess the performance of all methods on 8 NVIDI A100 GPUs.

Table 8: CNN Architecture

| Layer Type | Parameter Configuration |
| --- | --- |
| Conv2D | Input channels 3, output channels 32, kernel size 3×3, padding 1 |
| ReLU | Activation function |
| MaxPool2D | Pooling kernel 2×2, stride 2 |
| Conv2D | Input channels 32, output channels 64, kernel size 3×3, padding 1 |
| ReLU | Activation function |
| MaxPool2D | Pooling kernel 2×2, stride 2 |
| Flatten | Flatten to 64×8×8 vector |
| Linear | Input dimension 4096 (64×8×8), output dimension 128 |
| ReLU | Activation function |
| Linear | Input dimension 128, output dimension 10 |

**Evaluation Metrics.** For the CIFAR-10 dataset, we evaluate the model performance using six standard metrics: accuracy, precision, recall, F1-score, iterations, and wall-clock time. For the WikiText dataset, we evaluate the performance using four standard metrics: accuracy, perplexity, iterations, and wall-clock time. These metrics collectively assess both the effectiveness and efficiency of our PASO. For evaluation of the wall-clock time, we use the *torch.cuda.Event*[6] method provided by Pytorch (Paszke et al., 2019).

**Algorithms**. We accelerate three widely used optimizers: SGD, Adam, and AdamW. We refer their parallel variants as ParaSGD, ParaAdam, and ParaAdamW, respectively.

## D.2 EXPERIMENT RESULTS

### D.2.1 EMPIRICAL VALIDATION OF OPTIMIZATION TRAJECTORY EQUIVALENCE

The objective of this study is to verify that the PASO algorithm faithfully reproduces the optimization path of various standard sequential optimizers. The experimental design is as follows:

- **Model and Task:** We train a CNN on the CIFAR-10 dataset. The total iterations is 10000 GD steps.

- **Optimizer Comparison:** For each base optimizer (SGD, Adam, AdamW), we conduct two parallel training procedures: one using the standard sequential optimizer and another using its PASO-enhanced version. Critically, all training runs commence from an identical set of randomly initialized weights, learning rate, and batch size to ensure a fair comparison.

- **Evaluation Metric:** At each training step $t$, we compute the squared L2 norm of the difference between the model weight vectors produced by the two methods, defined as:

$$d^t = ||w^t_{\text{PASO}} - w^t_{\text{Sequential}}||^2$$

where $w^t_{\text{PASO}}$ and $w^t_{\text{Sequential}}$ represent the model weights obtained by the PASO variant and its standard sequential counterpart at step $t$, respectively. This metric, $d^t$, quantifies the instantaneous deviation between the two optimization trajectories in the parameter space. To provide a comprehensive assessment, we report the mean and variance of $d^t$ across the entire training process for each optimizer.

---

[3] https://github.com/openai/gpt-2

[4] https://huggingface.co/meta-llama/Llama-3.2-1B

[5] https://www.salesforce.com/blog/the-wikitext-long-term-dependency-language-modeling-dataset

[6] https://pytorch.org/docs/stable/generated/torch.cuda.Event.html

**Results and Analysis.** The statistical summary of the trajectory divergence $d^t$ for all optimizers over the complete training process is presented in Table 9.

Table 9: Statistical summary of trajectory divergence ($d^t$) between PASO and sequential optimizers.

| Optimizer | Mean of $d^t$ | Variance of $d^t$ |
|---|---|---|
| SGD and SGD + PASO | $3.14 \times 10^{-6}$ | $6.71 \times 10^{-12}$ |
| Adam and Adam + PASO | $3.56 \times 10^{-3}$ | $5.75 \times 10^{-6}$ |
| AdamW and AdamW + PASO | $3.38 \times 10^{-3}$ | $4.70 \times 10^{-6}$ |

The results demonstrate that for all three optimizers, the mean and variance of the divergence $d^t$ remain exceptionally small throughout the training process. The consistently minimal values across all optimizers empirically confirm that PASO faithfully reproduces the optimization trajectory of the standard sequential optimizer, regardless of the specific optimization algorithm employed. This high-fidelity replication ensures that the convergence properties and final solution quality of the original optimizer are preserved. Note that while the average L2 norm for Adam and AdamW appear larger than SGD's, they remain highly insignificant when considered in context. For a model with millions of parameters, an average squared L2 norm difference on the order of $10^{-3}$ corresponds to an extremely small per-parameter discrepancy. For example, for a model with $n \approx 5 \times 10^6$ parameters, this corresponds to a *root mean squared error (RMSE)* per parameter of approximately $\sqrt{3.56 \times 10^{-3}/5 \times 10^6} \approx 2.7 \times 10^{-5}$. Consequently, the trajectories of PASO and sequential optimizers are functionally equivalent.

### D.2.2 Language Modeling Task

**Complete Training Process.** Figure 6 presents the loss, accuracy, and perplexity curves of ParaSGD, ParaAdam, and ParaAdamW, respectively. We can see that ParaSGD, ParaAdam, and ParaAdamW achieve faster convergence across iterations. For example, for our ParaSGD method (Figure 2a, from top to bottom), both the training loss and perplexity decrease rapidly after approximately 200 iterations, while the loss and perplexity of SGD drop slowly and stabilizes around 1000. The training accuracy of ParaSGD increases quickly, reaching close to 80% by the $200th$ iteration. This suggests that our PASO is effective in improving the training efficiency.

### D.2.3 Image Classification Task

**Complete Training Process.** Figure 7 shows the training loss and accuracy trajectories for ParaSGD, ParaAdam, and ParaAdamW. All three methods demonstrate accelerated convergence compared to their baseline counterparts. Particularly notable is ParaAdamW (Figure 3c, top to bottom), where the training loss exhibits a sharp decline after approximately 3.5k iterations - in contrast to AdamW's gradual reduction that only stabilizes around 30k iterations. Furthermore, ParaAdamW achieves a training accuracy of nearly 82% by the 3.5k-th iteration, significantly outperforming AdamW's 69% accuracy at 30k iterations. These results demonstrate that our PASO framework effectively enhances both training efficiency and model performance metrics.

### D.2.4 The Impact of Hyperparameters

**Impact of Tolerance $\delta$ and EMA Decay Rate $\lambda$.** The Fig. 8 illustrates the impact of different tolerance ($\delta$) and EMA decay rate ($\lambda$) on model performance. The results show that different combinations of $\delta$ and $\lambda$ achieve a speedup of $4.61\times$ (13000 v.s. 60000) to $4.81\times$ (12450 v.s. 60000) while maintaining the similar model quality as Adam. In addition, the interplay between $\delta$ and $\lambda$ highlights a trade-off: aggressive smoothing ($\lambda \uparrow$) with loose tolerance ($\delta \uparrow$) may reduce computational effort, while finer tolerance ($\delta \downarrow$) with moderate $\lambda$ could enhance model quality at the expense of convergence speed.

In summary, these new parameters do not require extensive tuning and are quite intuitive in selection:

- **Tolerance ($\delta$)**: This parameter controls the convergence precision of the fixed-point iteration. Manually setting this could be tedious. For this reason, we employ an adaptive tolerance schedule. The tolerance starts loose and automatically tightens as training progresses. This makes the method robust and largely removes $\delta$ from the list of parameters requiring manual tuning.

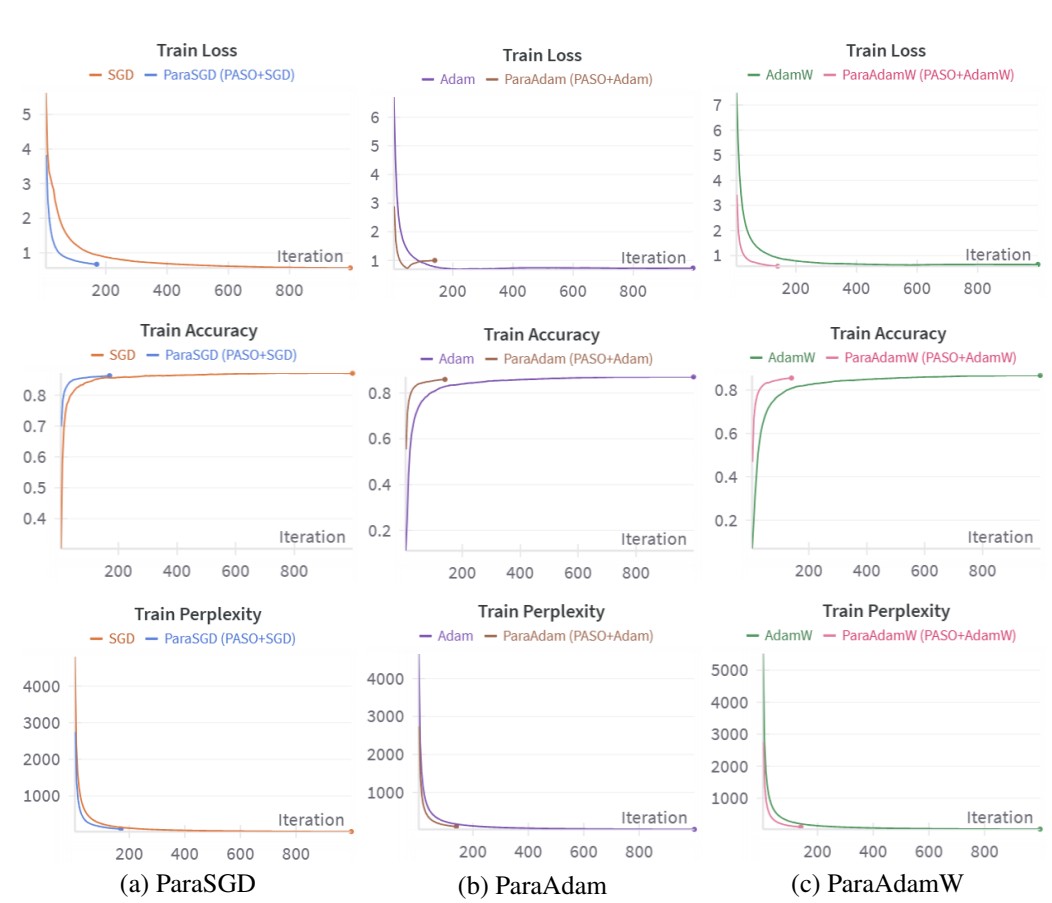

Figure 6: The loss, accuracy, and perplexity curve of WikiText Task

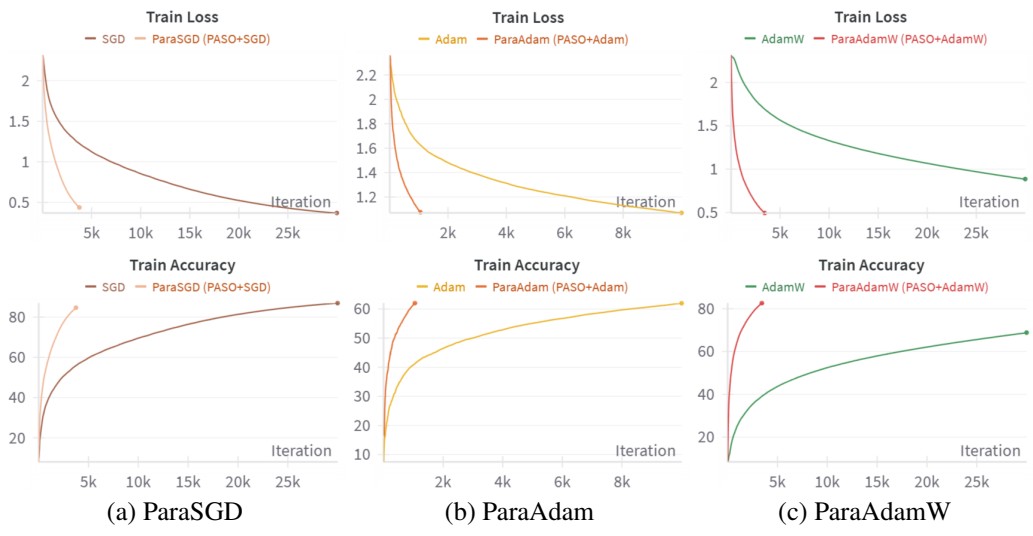

Figure 7: The loss and accuracy curve of CIFAR10 Task

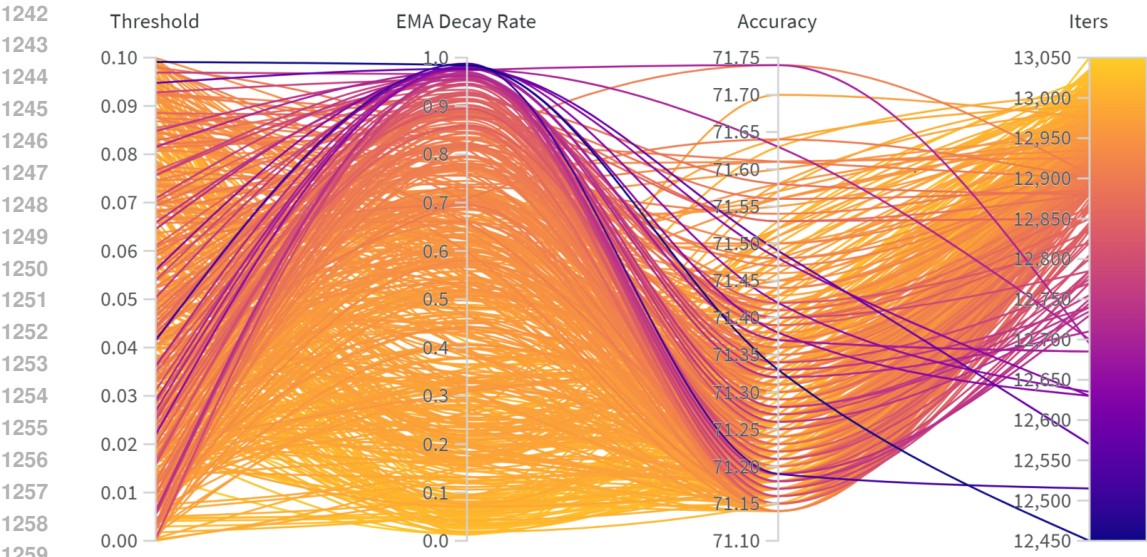

Figure 8: The impact of $\delta$ and $\lambda$ over CIFAR-10 by running 1200 experiments. We use PASO with $p = 7$ to accelerate Adam with 60000 steps. Darker lines indicate runs with fewer iterations.

- **EMA Decay Rate** ($\lambda$): This is used within our adaptive tolerance schedule. Like most EMA parameters in deep learning (e.g., in batch normalization or Adam), it is not highly sensitive.
- **Window Size** ($p$): This is less of a hyperparameter and more of a hardware configuration parameter. For good efficiency, $p$ can be simply set as the number of available processors.

## E  DETAILED COMPARATIVE ANALYSIS

In this section, we provide the detailed derivations and analyses of computational cost, memory footprint, and speedup ratios for sequential SGD and various parallel training methods, as summarized in Table 1.

### E.1  COMPUTATIONAL COST AND MEMORY FOOTPRINT ANALYSIS

To quantify the overhead of PASO, let $T$ be the total number of training steps for a standard sequential method. PASO converges in $K$ iterations, with each iteration performing $p$ parallel gradient computations (where $p$ is the window size). The total maximum number of gradient computations is therefore $p \times K$. Since the sliding window size $p$ will gradually decrease at the end of the convergence, the practical number for gradients computations (we denote it as $G$) is less than $pK$. We define the *computational cost ratio* $m$ as the ratio of PASO's total gradient computations to that of the sequential method:

$$m = \frac{pK}{T}$$

Empirically, as shown in Figure 9, our experiments for $T = 10000$ demonstrate that $m$ remains close to 1 and does not exceed 1.5 across various window sizes. This indicates that PASO introduces minimal computational overhead.

In terms of memory, PASO requires storing only one model and one optimizer state per device. This is identical to the requirements of sequential, model, and pipeline parallelism. It is also significantly more memory-efficient than data parallelism, where the storage for optimizer states typically scales with the number of devices $N$.

### E.2  SPEEDUP RATIO ANALYSIS

In this section, we provide a detailed derivation of the speedup ratios for sequential SGD and various parallel training methods, as summarized in Table 1.

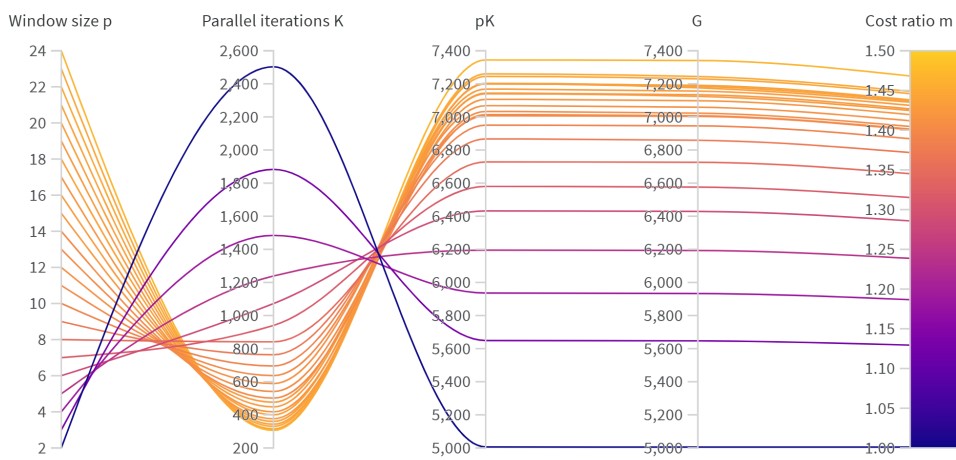

Figure 9: Empirical evaluation of the computational cost ratio $m = pK/T$ for $T = 10000$ across different window sizes $p$. Since the sliding window size $p$ will gradually decrease at the end of the convergence, the actual total number of gradient computations (we denote it as $G$) is marginally less than $pK$. The ratio remains close to 1, indicating minimal computational overhead.

### E.2.1 DEFINITIONS AND ASSUMPTIONS

For a clear and consistent analysis, we define the following notations:

- $N$: The number of GPUs, assumed to have identical compute capabilities.
- $T$: The total number of iterations (steps) required for a model to converge using sequential SGD.
- $t_{\text{comp}}$: The time required for the computation within one SGD step on a single GPU. For simplicity, we normalize this to $t_{\text{step}}$ in some contexts.
- $t_{\text{comm}}$: The time required for necessary communication (e.g., synchronization) per parallel step.
- $\alpha \triangleq t_{\text{comm}}/t_{\text{comp}}$: The communication-to-computation ratio, a critical factor in parallel efficiency.

The **speedup ratio** $S$ for any parallel method is defined as the ratio of the total time taken by sequential SGD to the time taken by the parallel method:

$$S = \frac{T_{\text{sequential}}}{T_{\text{parallel}}}$$

### E.2.2 BASELINE: SEQUENTIAL SGD

The total time for sequential SGD is the product of the number of iterations and the time per iteration.

$$T_{\text{sequential}} = T \times t_{\text{comp}}$$

By definition, its speedup ratio is $S_{\text{sequential}} = 1$.

### E.2.3 DATA-PARALLEL TRAINING

In synchronous data parallelism, the computation for each step is divided across $N$ GPUs, but a communication step (e.g., AllReduce) is required to synchronize gradients. The time for one parallel step is $\left(\frac{t_{\text{comp}}}{N} + t_{\text{comm}}\right)$. The total time over $T$ iterations is:

$$T_{\text{data}} = T \times \left(\frac{t_{\text{comp}}}{N} + t_{\text{comm}}\right)$$

The speedup ratio is therefore:

$$S_{\text{data}} = \frac{T \cdot t_{\text{comp}}}{T\left(\frac{t_{\text{comp}}}{N} + t_{\text{comm}}\right)} = \frac{t_{\text{comp}}}{\frac{t_{\text{comp}}}{N} + \alpha \cdot t_{\text{comp}}} = \frac{1}{\frac{1}{N} + \alpha} = \frac{N}{1 + \alpha N}$$

In the communication-bound limit ($N \to \infty$), the speedup is capped at $S_{\text{data}} \to 1/\alpha$.

### E.2.4 MODEL-PARALLEL AND PIPELINE-PARALLEL TRAINING

For both model and pipeline parallelism, assuming perfect load balancing and ignoring initial pipeline-filling latency for large $T$, the computation is similarly distributed. The model is partitioned across $N$ devices, and each device computes its part in parallel, followed by communication of activations or gradients between devices. The total time can be approximated as:

$$T_{\text{model/pipeline}} \approx T \times \left(\frac{t_{\text{comp}}}{N} + t_{\text{comm}}\right)$$

This yields the same speedup ratio form as data parallelism:

$$S_{\text{model/pipeline}} = \frac{N}{1 + \alpha N}$$

Practical limitations such as load imbalance or pipeline bubble latency often result in a lower effective speedup.

### E.2.5 STEP-PARALLEL TRAINING (PASO)

PASO operates differently by parallelizing across training steps. We introduce three key parameters for its analysis:

- $p$: The window size, representing the number of gradient steps computed in parallel.
- $K$: The total number of PASO iterations required for convergence.
- $m$: The computational cost ratio, $m = \frac{pK}{T}$, where $pK$ is the total number of gradient computations performed by PASO. Our empirical results show $m \approx 1$.

In each of the $K$ iterations, $p$ gradients are computed in parallel across $N$ devices. The computation time per iteration is $\frac{p \cdot t_{\text{comp}}}{N}$, followed by a single communication phase $t_{\text{comm}}$. The total time for PASO is:

$$T_{\text{PASO}} = K \times \left(\frac{p \cdot t_{\text{comp}}}{N} + t_{\text{comm}}\right)$$

To compare this with sequential training over $T$ steps, we substitute $K = \frac{mT}{p}$:

$$T_{\text{PASO}} = \frac{mT}{p}\left(\frac{p \cdot t_{\text{comp}}}{N} + t_{\text{comm}}\right) = mT\left(\frac{t_{\text{comp}}}{N} + \frac{t_{\text{comm}}}{p}\right)$$

The speedup ratio for PASO is then:

$$S_{\text{PASO}} = \frac{T \cdot t_{\text{comp}}}{mT\left(\frac{t_{\text{comp}}}{N} + \frac{t_{\text{comm}}}{p}\right)} = \frac{t_{\text{comp}}}{m\left(\frac{t_{\text{comp}}}{N} + \frac{\alpha \cdot t_{\text{comp}}}{p}\right)} = \frac{1}{m\left(\frac{1}{N} + \frac{\alpha}{p}\right)} = \frac{N}{m(1 + \alpha N/p)}$$

Since our experiments show $m \approx 1$ (see Figure 9), the speedup is approximately:

$$S_{\text{PASO}} \approx \frac{N}{1 + \alpha N/p}$$

As $p > 1$, it follows that $1 + \alpha N/p < 1 + \alpha N$, which confirms that $S_{\text{PASO}} > S_{\text{data/model/pipeline}}$. In the communication-bound limit ($N \to \infty$), the speedup is capped at $S_{\text{PASO}} \to p/(m\alpha)$, which is $p$ times higher than other methods.

Table 10: Summary of Notations

| Notation | Description |
|---|---|
| $T$ | Total number of gradient descent steps |
| $t$ | Current step index, $t \in \{0, 1, \ldots, T-1\}$ |
| $w_t$ | Model parameters at step $t$ |
| $\eta_t$ | Learning rate at step $t$ |
| $\zeta_t$ | Mini-batch of data used at step $t$ |
| $\mathcal{L}(w_t, \zeta_t)$ | Loss function evaluated at parameters $w_t$ with data $\zeta_t$ |
| $\nabla_{w_t} \mathcal{L}(w_t, \zeta_t)$ | Gradient of loss with respect to $w_t$ |
| $g_t(\cdot)$ | Update function specific to optimizer (SGD, Adam, etc.) |
| $r$ | The number of history weights used for existing autoregressive optimizers |
| $F_t(\cdot)$ | Nonlinear equation function at step $t$ |
| $\hat{w}_t^{(k)}$ | Estimated parameters at step $t$, iteration $k$ |
| $K$ | Number of parallel iterations |
| $p$ | Sliding window size for parallel computation |
| $\delta$ | Convergence tolerance threshold |
| $\lambda$ | Exponential moving average decay rate |
| $n$ | Dimension of model parameters |
| $L$ | Lipschitz constant for gradients |
| $M$ | Bound on gradient norm |
| $\epsilon$ | Small constant for numerical stability |
| $\beta_1, \beta_2$ | Exponential decay rates for optimizer with momentum |

## F    NOTATION SUMMARY

## G    UPDATE RULES FOR VARIOUS OPTIMIZERS IN DEFINITION 2

**Adam Optimizer**. At each iteration $t$, Adam computes the gradient of the loss function $\mathcal{L}(w_t, \zeta_t)$ over the mini-batch $\zeta_t$. It then updates two key quantities: the first moment $m_t$, which captures the momentum of the gradients, and the second moment $v_t$, which estimates the variability of the gradients. These updates are governed by exponential moving averages:

$$m_t = \beta_1 m_{t-1} + (1-\beta_1)\nabla_{w_{t-1}}\mathcal{L}(w_{t-1}, \zeta_{t-1}), v_t = \beta_2 v_{t-1} + (1-\beta_2)(\nabla_{w_{t-1}}\mathcal{L}(w_{t-1}, \zeta_{t-1}))^2, \tag{11}$$

where $\beta_1$ and $\beta_2$ are hyperparameters controlling the decay rates of the moving averages. Adam applies bias correction to the moments:

$$\hat{m}_t = \frac{m_t}{1-\beta_1^t}, \hat{v}_t = \frac{v_t}{1-\beta_2^t}. \tag{12}$$

The model parameters $w$ are then updated using the following rule:

$$w_t = w_{t-1} - \eta_{t-1}\frac{\hat{m}_t}{\sqrt{\hat{v}_t} + \epsilon}, \tag{13}$$

where the division is defined as the Hadamard division, and $\epsilon$ is a small constant.

For Adam , reformulating Eq. (11) produces the formulas of their general terms:

$$m_t = (1-\beta_1)\sum_{\tau=0}^{t-1}\beta_1^{t-1-\tau}\nabla_{w_\tau}\mathcal{L}(w_\tau, \zeta_\tau), v_t = (1-\beta_2)\sum_{\tau=0}^{t-1}\beta_2^{t-1-\tau}\left(\nabla_{w_\tau}\mathcal{L}(w_\tau, \zeta_\tau)\right)^2. \tag{14}$$

Through the combination of Eq. (12), Eq. (13), and Eq. (14), we derive $g_{t-1}$ with $r = t$ for Adam as follows:

$$g_{t-1}(w_{t-1}, \cdots, w_0; \zeta_{t-1}, \cdots, \zeta_0) = \frac{\frac{1-\beta_1}{1-\beta_1^t}\sum_{\tau=0}^{t-1}\beta_1^{t-1-\tau}\nabla_{w_\tau}\mathcal{L}(w_\tau, \zeta_\tau)}{\sqrt{\frac{(1-\beta_2)\sum_{\tau=0}^{t-1}\beta_2^{t-1-\tau}(\nabla_{w_\tau}\mathcal{L}(w_\tau, \zeta_\tau))^2}{1-\beta_2^t}} + \epsilon}. \tag{15}$$

**AdamW Optimizer**. The explicit form of $g_\tau$ for AdamW is derived by decoupling weight decay from the Adam update rule. Let $r = \tau$, then:

$$g_\tau(w_\tau, \ldots, w_0; \zeta_\tau, \ldots, \zeta_0) = \frac{\frac{1-\beta_1}{1-\beta_1^{\tau+1}} \sum_{k=0}^{\tau} \beta_1^{\tau-k} \nabla_{w_k} \mathcal{L}(w_k, \zeta_k)}{\sqrt{\frac{(1-\beta_2)}{1-\beta_2^{\tau+1}} \sum_{k=0}^{\tau} \beta_2^{\tau-k} (\nabla_{w_k} \mathcal{L}(w_k, \zeta_k))^2 + \epsilon}} + \lambda w_\tau,$$

where $\lambda$ is the weight decay coefficient. The term $\lambda w_\tau$ is explicitly added to the original Adam update, independent of gradient history.

**Adagrad Optimizer**. For Adagrad, the update function $g_\tau$ is defined using the explicit sum of squared gradients up to iteration $\tau$:

$$g_\tau(w_\tau, \ldots, w_0; \zeta_\tau, \ldots, \zeta_0) = \frac{\nabla_{w_\tau} \mathcal{L}(w_\tau, \zeta_\tau)}{\sqrt{\sum_{k=0}^{\tau} (\nabla_{w_k} \mathcal{L}(w_k, \zeta_k))^2 + \epsilon}}.$$

Here, the denominator is the square root of the *non-decaying cumulative sum* of all historical squared gradients. $\epsilon$ is a small constant added for numerical stability.

**SAM Optimizer**. The explicit form of $g_\tau$ for SAM (Sharpness-Aware Minimization) involves computing the gradient at a perturbed point. Here in SAM $r = 1$, then:

$$g_\tau(w_\tau; \zeta_\tau) = \nabla_{w_\tau + \varepsilon_\tau} \mathcal{L}(w_\tau + \varepsilon_\tau, \zeta_\tau),$$

where the perturbation $\varepsilon_\tau$ is defined as:

$$\varepsilon_\tau = \rho \cdot \frac{\nabla_{w_\tau} \mathcal{L}(w_\tau, \zeta_\tau)}{\|\nabla_{w_\tau} \mathcal{L}(w_\tau, \zeta_\tau)\|_2 + \delta}.$$

Substituting the expression for $\varepsilon_\tau$ into the gradient formula, we get:

$$g_\tau(w_\tau; \zeta_\tau) = \nabla_{w_\tau} \mathcal{L}\left(w_\tau + \rho \cdot \frac{\nabla_{w_\tau} \mathcal{L}(w_\tau, \zeta_\tau)}{\|\nabla_{w_\tau} \mathcal{L}(w_\tau, \zeta_\tau)\|_2 + \epsilon}, \zeta_\tau\right).$$

This formulation explicitly shows SAM computes the gradient at a point that is perturbed in the direction of steepest ascent within a neighborhood of radius $\rho$, seeking parameters that are robust to adversarial perturbations. Here, $\rho$ is the perturbation radius that controls the magnitude of the perturbation, and $\epsilon$ is a small constant added for numerical stability.

# H  PROOF OF PROPOSITION 1

We begin by showing that the function $F_{t-1}(\hat{w}_0, \cdots, \hat{w}_{t-1}; \zeta_0, \ldots, \zeta_{t-1})$ has a unique set of solutions. Assume there exist two distinct solutions, $A_0, \cdots, A_T$ and $B_0, \cdots, B_T$. For all $t \in \{0, T\}$, these solutions must satisfy:

$$\begin{cases} A_t = F_{t-1}(A_0, \cdots, A_{t-1}; \zeta_0, \ldots, \zeta_{t-1}) \\ B_t = F_{t-1}(B_0, \cdots, B_{t-1}; \zeta_0, \ldots, \zeta_{t-1}). \end{cases} \tag{16}$$

By mathematical induction, suppose $A_\tau = B_\tau$ for $0 \le \tau \le t$. Then,

$$A_{t+1} = F_t(A_0, \cdots, A_t; \zeta_0, \ldots, \zeta_t) = F_t(B_0, \cdots, B_t; \zeta_0, \ldots, \zeta_t) = B_{t+1}, \tag{17}$$

which implies $A_0, \cdots, A_T$ and $B_0, \cdots, B_T$ are identical. Thus, the solution is unique.

Next, we show that the solution of the triangular nonlinear equation system is an unbiased estimator for the autoregressive gradient descent process. From Eq. (3), the expectation of the autoregressive GD process is:

$$E[w_t] = E[F_{t-1}(w_0, \cdots, w_{t-1}; \zeta_0, \cdots, \zeta_{t-1})]$$

$$= E[w_0] - \sum_{\tau=0}^{t-1} E\left[\eta_\tau g_\tau(w_\tau, \ldots, w_{\tau-r+1}; \zeta_\tau, \ldots, \zeta_{\tau-r+1})\right]. \tag{18}$$

For the triangular NE system, we have:

$$E[\hat{w}_t] = E[F_{t-1}(\hat{w}_0, \cdots, \hat{w}_{t-1}; \zeta_0, \ldots, \zeta_{t-1})]$$

$$= E[\hat{w}_0] - E\left[\sum_{\tau=0}^{t-1} \eta_\tau g_\tau(\hat{w}_\tau, \ldots, \hat{w}_{\tau-r+1}; \zeta_\tau, \ldots, \zeta_{\tau-r+1})\right]. \tag{19}$$

Since $w_0$ and $\hat{w}_0$ follow the same distribution, and $\eta_t$ and the mini-batches $\zeta_t$ are identical across all time steps, it follows that:

$$E[\hat{w}_t] = E[w_t], \quad \forall\, 0 \le t \le T.$$

## I    PROOF OF CONVERGENCE FOR FIXED-POINT ITERATION IN PROPOSITION 2

### I.1    ASSUMPTIONS AND LEMMAS

To give the proof, we first state the underlying assumptions and lemmas used:

**Assumption 1.** *The gradient $\nabla_{w_\tau}\mathcal{L}(w_\tau, \zeta_\tau)$ is L-Lipschitz continuous:*

$$\|\nabla_{w_\tau}\mathcal{L}(w_\tau, \zeta_\tau) - \nabla_{w_\tau}\mathcal{L}(x_\tau, \zeta_\tau)\| \le L\|w_\tau - x_\tau\|. \tag{20}$$

**Assumption 2.** *The gradient norm is bounded:*

$$\|\nabla_{w_\tau}\mathcal{L}(w_\tau, \zeta_\tau)\| \le M. \tag{21}$$

*This implies bounded model weights $w_\tau$. For example, consider the simply quadratic loss $\mathcal{L}(w, \zeta) = w^2$ (with $w \in \mathbb{R}$ independent of $\zeta$). Here:*

$$\nabla_w \mathcal{L} = 2w, \quad so \quad |\nabla_w \mathcal{L}| = |2w|.$$

*The bounded gradient condition $|2w| \le M$ directly implies $|w| \le M/2$, proving $w$ is constrained to a compact set.*

**Lemma 1.** *If $U, V \in \mathbb{R}^{n \times t}$ satisfy $U_{ij}, V_{ij} \ge \mu$ for all $i, j$, then*

$$\|\sqrt{U} - \sqrt{V}\|_F \ \le\ \frac{1}{2\sqrt{\mu}}\, \|U - V\|_F,$$

*where the square-root is taken element-wise.*

*Proof.* For any scalars $a, b \ge \mu > 0$,

$$\left|\sqrt{a} - \sqrt{b}\right| = \frac{|a - b|}{\sqrt{a} + \sqrt{b}} \ \le\ \frac{|a - b|}{2\sqrt{\mu}},$$

because $\sqrt{a} + \sqrt{b} \ge 2\sqrt{\mu}$.

Applying this entrywise with $a = U_{ij}$ and $b = V_{ij}$ yields

$$\left|\sqrt{U_{ij}} - \sqrt{V_{ij}}\right| \ \le\ \frac{1}{2\sqrt{\mu}}\, |U_{ij} - V_{ij}| \qquad (\forall\, i, j).$$

Squaring and summing over $(i, j)$,

$$\sum_{i,j}\left(\sqrt{U_{ij}} - \sqrt{V_{ij}}\right)^2 \ \le\ \frac{1}{4\mu}\sum_{i,j}(U_{ij} - V_{ij})^2.$$

The left–hand side equals $\|\sqrt{U} - \sqrt{V}\|_F^2$ and the right–hand side equals $\frac{1}{4\mu}\|U - V\|_F^2$.

Taking square roots gives

$$\|\sqrt{U} - \sqrt{V}\|_F \ \le\ \frac{1}{2\sqrt{\mu}}\, \|U - V\|_F,$$

$\square$

## I.2 PROBLEM RESTATEMENT

**Notation.** We denote the collection of weights up to time $\tau$ as $W_\tau = [\hat{w}_0, \ldots, \hat{w}_\tau]$ and note $W_{T-1} = [\hat{w}_0, \ldots, \hat{w}_{T-1}]$ as $W$. The norm $\|\cdot\|$ is the Frobenius norm. For model weights $w \in \mathbb{R}^n$ with $n > 1$, multiplication and division are element-wise (Hadamard product and division).

**Definition 3** (Iterative Mapping). *Let the iterative mapping $\mathcal{H} : \mathbb{R}^{n \times T} \to \mathbb{R}^{n \times T}$ (T components) be defined as follows for a sequence of model weights $W = [\hat{w}_0, \hat{w}_1, \ldots, \hat{w}_{T-1}]$:*

$$\mathcal{H}(\hat{w}_0, \cdots, \hat{w}_{T-1}) = \begin{cases} \hat{w}_0 = w_0^{seq}, \\ F_0(\hat{w}_0; \zeta_0), \\ F_1(\hat{w}_0, \hat{w}_1; \zeta_0, \zeta_1), \\ \cdots, \\ F_{T-1}(\hat{w}_0, \cdots, \hat{w}_{T-1}; \zeta_0, \ldots, \zeta_{T-1}), \end{cases} \tag{22}$$

*where $w_0^{seq}$ denotes the initialized model for the sequential gradient descent and each sub-mapping $F_{t-1}$ is of the form:*

$$F_{t-1}(\hat{w}_0, \cdots, \hat{w}_{t-1}) = \hat{w}_0 - \sum_{\tau=0}^{t-1} \eta_\tau g_\tau(\hat{w}_\tau, \ldots, \hat{w}_0). \tag{23}$$

*The fixed-point iteration is thus defined by the sequence $W^k = \mathcal{H}(W^{k-1})$.*

**Definition 4** (Autoregressive Gradient Descent Trajectory). *The target fixed point, denoted by $W^{seq} = [w_0^{seq}, w_1^{seq}, \ldots, w_{T-1}^{seq}]$, is the trajectory generated by autoregressive gradient descent:*

$$w_0^{seq} = initial\ model\ weight \tag{24}$$

$$w_t^{seq} = w_0^{seq} - \sum_{\tau=0}^{t-1} \eta_\tau g_\tau(w_\tau^{seq}, \ldots, w_0^{seq}) \quad for\ t \geq 1 \tag{25}$$

*It is straightforward to see that $W^{seq}$ is a fixed point of $\mathcal{H}$, since $\mathcal{H}(W^{seq}) = W^{seq}$.*

## I.3 OBJECTIVES

We aim to prove two key properties of this iterative process:

1. **Convergence:** The fixed-point iteration $W^k = \mathcal{H}(W^{k-1})$ converges to the unique fixed point $W^{seq}$, which corresponds to the trajectory of autoregressive gradient descent.

2. **Finite Convergence Steps:** In the worst-case scenario, the number of iterations $K$ required for convergence ($W^K = W^{seq}$) is at most $T$.

## I.4 PROOF OF CONVERGENCE (OBJECTIVE 1)

We will prove by mathematical induction on the time step $t$ that for each $t \in \{0, \ldots, T-1\}$, the sequence of iterates $\{\hat{w}_t^k\}_{k=1}^{\infty}$ converges to $w_t^{seq}$.

*Proof.* Let $W^k = [\hat{w}_0^k, \ldots, \hat{w}_{T-1}^k]$ be the iterates at step $k$. From the definition of $\mathcal{H}$, we have:

$$\hat{w}_0^k = w_0^{seq} \tag{26}$$

$$\hat{w}_t^k = \hat{w}_0^{k-1} - \sum_{\tau=0}^{t-1} \eta_\tau g_\tau(\hat{w}_\tau^{k-1}, \ldots, \hat{w}_0^{k-1}) \quad \text{for } t \geq 1 \tag{27}$$

**Base Case ($t = 0$):** From the definition of $\mathcal{H}$, $\hat{w}_0^k = w_0^{seq}$ for all $k \geq 1$. Thus,

$$\lim_{k \to \infty} \left\| \hat{w}_0^k - w_0^{seq} \right\|_F = 0$$

The base case holds trivially.

**Inductive Hypothesis:** Assume for a given $t \geq 0$ that for all $\tau \in \{0, \ldots, t\}$, we have:

$$\lim_{k \to \infty} \left\| \hat{w}_\tau^k - w_\tau^{seq} \right\|_F = 0$$

**Inductive Step:** We must show that the statement holds for $t+1$, i.e., $\lim_{k \to \infty} \left\| \hat{w}_{t+1}^k - w_{t+1}^{seq} \right\|_F = 0$.

The iterate $\hat{w}_{t+1}^k$ and the target $w_{t+1}^{seq}$ are given by:

$$\hat{w}_{t+1}^k = \hat{w}_0^{k-1} - \sum_{\tau=0}^{t} \eta_\tau g_\tau(W_\tau^{k-1})$$

$$w_{t+1}^{seq} = w_0^{seq} - \sum_{\tau=0}^{t} \eta_\tau g_\tau(W_\tau^{seq})$$

Since $\hat{w}_0^{k-1} = w_0^{seq}$ for $k - 1 \geq 1$, the difference is:

$$\hat{w}_{t+1}^k - w_{t+1}^{seq} = \sum_{\tau=0}^{t} \eta_\tau \left( g_\tau(W_\tau^{seq}) - g_\tau(W_\tau^{k-1}) \right)$$

Taking the norm and applying the triangle inequality:

$$\left\| \hat{w}_{t+1}^k - w_{t+1}^{seq} \right\|_F \leq \sum_{\tau=0}^{t} \eta_\tau \left\| g_\tau(W_\tau^{k-1}) - g_\tau(W_\tau^{seq}) \right\|_F$$

From Appendix I.6, I.7, and I.8. we can know that the gradient function $g_\tau$ for various optimizers is upper bounded with respect to its arguments. Denote uniformly by these boundaries $C$, we have:

$$\left\| \hat{w}_{t+1}^k - w_{t+1}^{seq} \right\|_F \leq \sum_{\tau=0}^{t} \eta_\tau C \left\| W_\tau^{k-1} - W_\tau^{seq} \right\|_F$$

By the inductive hypothesis, for each $\tau \in \{0, \ldots, t\}$, every component of $W_\tau^{k-1}$ converges to the corresponding component of $W_\tau^{seq}$ as $k \to \infty$. This implies that:

$$\lim_{k \to \infty} \left\| W_\tau^{k-1} - W_\tau^{seq} \right\|_F = \lim_{k \to \infty} \left( \sum_{j=0}^{\tau} \left\| \hat{w}_j^{k-1} - w_j^{seq} \right\|_F^2 \right)^{1/2} = 0$$

Since the sum on the right-hand side is a finite sum of terms each converging to zero, the entire expression converges to zero:

$$\lim_{k \to \infty} \left\| \hat{w}_{t+1}^k - w_{t+1}^{seq} \right\|_F \leq \sum_{\tau=0}^{t} \eta_\tau C \cdot 0 = 0$$

As the norm is non-negative, we conclude $\lim_{k \to \infty} \left\| \hat{w}_{t+1}^k - w_{t+1}^{seq} \right\|_F = 0$. This completes the inductive step.

By the principle of mathematical induction, $\hat{w}_t^k \to w_t^{seq}$ for all $t \in \{0, \ldots, T - 1\}$. Therefore, the iteration $W^k = \mathcal{H}(W^{k-1})$ converges to $W^{seq}$. $\qquad\square$

### I.5 PROOF OF CONVERGENCE STEPS (OBJECTIVE 2)

We now prove a stronger result: in worst-case scenario, the fixed-point iteration converges to the exact fixed point $W^{seq}$ in at most $T$ iterations.

**Worst-Case Scenario Analysis.** The structure of the mapping $\mathcal{H}$ imposes a causal dependency: the calculation of $\hat{w}_t^k$ depends only on the components $\hat{w}_0^{k-1}, \ldots, \hat{w}_{t-1}^{k-1}$ from the previous iteration. The initial models for the fixed-point iteration and the autoregressive gradient descent are identical at $t = 0$ ($\hat{w}_0^k = w_0^{seq}$). Consequently, convergence cannot occur "out of order". The component $\hat{w}_1$ can only converge after $\hat{w}_0$ has, $\hat{w}_2$ can only converge after $\hat{w}_0$ and $\hat{w}_1$ have, and so on.

The worst-case scenario occurs when each iteration $k$ can only ensure the convergence of one component, leading to the convergence proceeding sequentially, one component at a time. This sequential "locking-in" of the correct values is equivalent in its step-by-step nature to the autoregressive gradient descent. We will formalize this intuition below.

*Proof.* We will prove by induction on the component index $t$ the statement $P(t)$:

$$P(t): \quad \hat{w}_t^k = w_t^{seq} \quad \text{for all } k \geq t + 1.$$

**Base Case** ($t = 0$): We must prove $P(0)$: $\hat{w}_0^k = w_0^{seq}$ for all $k \geq 1$. By the definition of $\mathcal{H}$ in Eq. (22), $\hat{w}_0^k$ is set to $w_0^{seq}$ for every iteration $k \geq 1$. The base case holds.

**Inductive Hypothesis:** Assume for some $t \geq 1$ that $P(\tau)$ holds for all $\tau \in \{0, 1, \ldots, t - 1\}$. This means for each such $\tau$:

$$\hat{w}_\tau^k = w_\tau^{seq} \quad \text{for all } k \geq \tau + 1.$$

**Inductive Step:** We must prove that $P(t)$ holds: $\hat{w}_t^k = w_t^{seq}$ for all $k \geq t + 1$.

Consider an arbitrary iteration $k$ such that $k \geq t + 1$. This implies $k - 1 \geq t$. The iterate $\hat{w}_t^k$ is defined as:

$$\hat{w}_t^k = \hat{w}_0^{k-1} - \sum_{\tau=0}^{t-1} \eta_\tau g_\tau(\hat{w}_\tau^{k-1}, \ldots, \hat{w}_0^{k-1}).$$

The arguments to the functions $g_\tau$ are the components of $W^{k-1}$. Let's examine an arbitrary component $\hat{w}_\tau^{k-1}$ in this expression, where $\tau \in \{0, 1, \ldots, t - 1\}$. From our condition on $k$, we have $k - 1 \geq t > \tau$, which implies $k - 1 \geq \tau + 1$.

According to our inductive hypothesis, since $k - 1 \geq \tau + 1$, each of these components has already converged to its final value:

$$\hat{w}_\tau^{k-1} = w_\tau^{seq} \quad \text{for each } \tau \in \{0, 1, \ldots, t - 1\}.$$

This demonstrates that for any iteration $k \geq t + 1$, all the inputs required to compute $\hat{w}_t^k$ have already stabilized to their fixed-point values at the preceding step, $k - 1$.

Substituting these converged values back into the expression for $\hat{w}_t^k$:

$$\hat{w}_t^k = w_0^{seq} - \sum_{\tau=0}^{t-1} \eta_\tau g_\tau(w_\tau^{seq}, \ldots, w_0^{seq}).$$

The right-hand side of this equation is precisely the definition of the target sequential weight $w_t^{seq}$. Therefore,

$$\hat{w}_t^k = w_t^{seq}.$$

Since our choice of $k \geq t + 1$ was arbitrary, this equality holds for all such $k$. This proves $P(t)$ and completes the inductive step.

**Conclusion on Iteration Count.** By induction, we have shown that $\hat{w}_t^k = w_t^{seq}$ for all $k \geq t + 1$. For the entire vector $W^k = [\hat{w}_0^k, \ldots, \hat{w}_{T-1}^k]$ to converge, every component must have converged. The last component to converge is $\hat{w}_{T-1}^k$. Applying our result for $t = T - 1$:

$$\hat{w}_{T-1}^k = w_{T-1}^{seq} \quad \text{for all } k \geq (T - 1) + 1 = T.$$

At iteration $k = T$, we have $T \geq t + 1$ for all $t \in \{0, \ldots, T - 1\}$. This implies that every component $\hat{w}_t^T$ has converged to $w_t^{seq}$. Thus, the entire vector has converged:

$$W^T = W^{seq}.$$

Therefore, the fixed-point iteration requires exactly $K = T$ iterations to converge to the fixed point in the worst case, and it remains there for all subsequent iterations. The number of iterations $K$ required does not exceed the number of autoregressive steps $T$. $\square$

## I.6 UPPER BOUND FOR THE DIFFERENCE OF $g_t$ IN SGD

For SGD, the update function $g_t$ takes the form:

$$g_t(w_t; \zeta_t) = \nabla_{w_t} \mathcal{L}(w_t, \zeta_t) \tag{28}$$

We aim to find an upper bound for $\|g_t(w_t) - g_t(x_t)\|$. By directly applying Assumption 1 (L-Lipschitz continuity), we get:

$$\|g_t(w_t) - g_t(x_t)\| = \|\nabla_{w_t} \mathcal{L}(w_t, \zeta_t) - \nabla_{x_t} \mathcal{L}(x_t, \zeta_t)\| \leq L\|w_t - x_t\| \tag{29}$$

Therefore, for SGD, the Lipschitz constant of the update function $g_t$ is $L$.

## I.7 UPPER BOUND FOR THE DIFFERENCE OF $g_t$ IN ADAM

**Notation.** We denote the collection of weights up to time $t$ as $W_\tau = [w_0, \ldots, w_t]$ and note $W_{T-1} = [w_0, \ldots, w_{T-1}]$ as $W$. Analogously, $X_t = [x_0, \ldots, x_t]$ and $X = [x_0, \ldots, x_{T-1}]$.

Our objective is to derive an upper bound for the difference $\|g_{t-1}(W_{t-1}) - g_{t-1}(X_{t-1})\|_F$ for any $W_{t-1}$ and $X_{t-1}$.

The function $g_{t-1}$ is defined as:

$$g_{t-1}(W_{t-1}) = \frac{A(W_{t-1})}{\sqrt{B(W_{t-1}) + \epsilon}} \tag{30}$$

where the division and square root are element-wise operations. The numerator $A(W_{t-1})$ and denominator component $B(W_{t-1})$ are defined as the bias-corrected first and second moment estimates:

$$A(W_{t-1}) = \frac{1 - \beta_1}{1 - \beta_1^t} \sum_{\tau=0}^{t-1} \beta_1^{t-1-\tau} \nabla_{w_\tau} \mathcal{L}(w_\tau, \zeta_\tau) \tag{31}$$

$$B(W_{t-1}) = \frac{1 - \beta_2}{1 - \beta_2^t} \sum_{\tau=0}^{t-1} \beta_2^{t-1-\tau} \left(\nabla_{w_\tau} \mathcal{L}(w_\tau, \zeta_\tau)\right)^2 \tag{32}$$

This proof relies on two standard assumptions:

**1. $L$-Lipschitz Gradient**: The gradient of the loss function is $L$-Lipschitz continuous, i.e., $\|\nabla\mathcal{L}(w) - \nabla\mathcal{L}(x)\|_F \leq L\|w - x\|_F$.

**2. Bounded Gradient Norm**: The Frobenius norm of the stochastic gradients is uniformly bounded by a constant $M$, i.e., $\|\nabla\mathcal{L}(w, \zeta)\|_F \leq M$.

For clarity, we will temporarily omit the subscript $t - 1$ from $W$ and $X$ within the derivation and re-introduce it in the final result. We begin by decomposing the difference $g(W) - g(X)$ by adding and subtracting an intermediate term:

$$g(W) - g(X) = \left(\frac{A(W) - A(X)}{\sqrt{B(W) + \epsilon}}\right) + \left(\frac{A(X)}{\sqrt{B(W) + \epsilon}} - \frac{A(X)}{\sqrt{B(X) + \epsilon}}\right) \tag{33}$$

This can be expressed using the element-wise Hadamard product ($\odot$) as:

$$g(W) - g(X) = (A(W) - A(X)) \odot \frac{1}{\sqrt{B(W) + \epsilon}} + A(X) \odot \left(\frac{1}{\sqrt{B(W) + \epsilon}} - \frac{1}{\sqrt{B(X) + \epsilon}}\right) \tag{34}$$

By applying the triangle inequality to the Frobenius norm, we get:

$$\|g(W) - g(X)\|_F \leq \left\|(A(W) - A(X)) \odot \frac{1}{\sqrt{B(W) + \epsilon}}\right\|_F + \left\|A(X) \odot \left(\frac{1}{\sqrt{B(W) + \epsilon}} - \frac{1}{\sqrt{B(X) + \epsilon}}\right)\right\|_F \tag{35}$$

Next, we use the property of the Hadamard product, $\|U \odot V\|_F \leq \|U\|_{\max}\|V\|_F$, where $\|U\|_{\max}$ is the maximum absolute value of any element in $U$. This yields our main inequality:

$$\|g(W) - g(X)\|_F \leq \left\|\frac{1}{\sqrt{B(W) + \epsilon}}\right\|_{\max} \|A(W) - A(X)\|_F + \|A(X)\|_{\max} \left\|\frac{1}{\sqrt{B(W) + \epsilon}} - \frac{1}{\sqrt{B(X) + \epsilon}}\right\|_F \tag{36}$$

We now bound the four terms in Eq. (36).

**1. Bound for $\|A(W_{t-1}) - A(X_{t-1})\|_F$**

From the definition in Eq. (31), we have:

$$A(W) - A(X) = \frac{1 - \beta_1}{1 - \beta_1^t} \sum_{\tau=0}^{t-1} \beta_1^{t-1-\tau} \left(\nabla_{w_\tau} \mathcal{L}(w_\tau, \zeta_\tau) - \nabla_{x_\tau} \mathcal{L}(x_\tau, \zeta_\tau)\right) \tag{37}$$

Taking the Frobenius norm and applying the triangle inequality, then using the $L$-Lipschitz assumption and the fact that $\|w_\tau - x_\tau\|_F \le \|W - X\|_F$:

$$\|A(W) - A(X)\|_F \le \frac{1 - \beta_1}{1 - \beta_1^t} \sum_{\tau=0}^{t-1} \beta_1^{t-1-\tau} \|\nabla_{w_\tau} \mathcal{L}(w_\tau, \zeta_\tau) - \nabla_{x_\tau} \mathcal{L}(x_\tau, \zeta_\tau)\|_F$$

$$\le \frac{1 - \beta_1}{1 - \beta_1^t} \sum_{\tau=0}^{t-1} \beta_1^{t-1-\tau} L \|w_\tau - x_\tau\|_F \tag{38}$$

$$\le L \|W - X\|_F \left( \frac{1 - \beta_1}{1 - \beta_1^t} \sum_{\tau=0}^{t-1} \beta_1^{t-1-\tau} \right)$$

The sum of the bias-correction weights is equal to one. Thus, we have:

$$\|A(W_{t-1}) - A(X_{t-1})\|_F \le L \|W_{t-1} - X_{t-1}\|_F \tag{39}$$

**2. Bound for** $\left\| \frac{1}{\sqrt{B(W)+\epsilon}} \right\|_{\max}$

Since each entry of $B(W)$ is a weighted average of squared gradients, $B_{ij}(W) \ge 0$ for all $i, j$. It follows that $\sqrt{B_{ij}(W) + \epsilon} \ge \sqrt{\epsilon}$. Taking the reciprocal gives the bound:

$$\left\| \frac{1}{\sqrt{B(W)+\epsilon}} \right\|_{\max} = \max_{i,j} \frac{1}{\sqrt{B_{ij}(W)+\epsilon}} \le \frac{1}{\sqrt{\epsilon}} \tag{40}$$

**3. Bound for** $\|A(X)\|_{\max}$

Given the bounded gradient assumption $\|\nabla \mathcal{L}\|_F \le M$, and since $\|\cdot\|_{\max} \le \|\cdot\|_F$, we have $\|\nabla \mathcal{L}\|_{\max} \le M$.

$$\|A(X)\|_{\max} \le \left\| \frac{1 - \beta_1}{1 - \beta_1^t} \sum_{\tau=0}^{t-1} \beta_1^{t-1-\tau} \nabla_{x_\tau} \mathcal{L}(x_\tau, \zeta_\tau) \right\|_{\max}$$

$$\le \frac{1 - \beta_1}{1 - \beta_1^t} \sum_{\tau=0}^{t-1} \beta_1^{t-1-\tau} \|\nabla_{x_\tau} \mathcal{L}\|_{\max} \le M \tag{41}$$

**4. Bound for** $\left\| \frac{1}{\sqrt{B(W)+\epsilon}} - \frac{1}{\sqrt{B(X)+\epsilon}} \right\|_F$

Let $u = B(W) + \epsilon$ and $v = B(X) + \epsilon$. We have:

$$\left\| \frac{1}{\sqrt{u}} - \frac{1}{\sqrt{v}} \right\|_F = \left\| \frac{\sqrt{v} - \sqrt{u}}{\sqrt{u}\sqrt{v}} \right\|_F \le \left\| \frac{1}{\sqrt{uv}} \right\|_{\max} \|\sqrt{v} - \sqrt{u}\|_F \le \frac{1}{\epsilon} \|\sqrt{v} - \sqrt{u}\|_F \tag{42}$$

The function $f(x) = \sqrt{x}$ is $\frac{1}{2\sqrt{\epsilon}}$-Lipschitz on $[\epsilon, \infty)$, which implies $\|\sqrt{v} - \sqrt{u}\|_F \le \frac{1}{2\sqrt{\epsilon}} \|v - u\|_F$ (see Lemma 1). Therefore:

$$\left\| \frac{1}{\sqrt{B(W)+\epsilon}} - \frac{1}{\sqrt{B(X)+\epsilon}} \right\|_F \le \frac{1}{2\epsilon^{3/2}} \|B(W) - B(X)\|_F \tag{43}$$

To complete this bound, we must bound $\|B(W) - B(X)\|_F$. From Eq. (32), we analyze the difference of squares term $(\nabla_{w_\tau} \mathcal{L})^2 - (\nabla_{x_\tau} \mathcal{L})^2 = (\nabla_{w_\tau} \mathcal{L} - \nabla_{x_\tau} \mathcal{L}) \odot (\nabla_{w_\tau} \mathcal{L} + \nabla_{x_\tau} \mathcal{L})$. Taking the norm:

$$\|(\nabla_{w_\tau} \mathcal{L})^2 - (\nabla_{x_\tau} \mathcal{L})^2\|_F \le \|\nabla_{w_\tau} \mathcal{L} - \nabla_{x_\tau} \mathcal{L}\|_F \cdot \|\nabla_{w_\tau} \mathcal{L} + \nabla_{x_\tau} \mathcal{L}\|_{\max}$$

$$\le (L \|w_\tau - x_\tau\|_F) \cdot (\|\nabla_{w_\tau} \mathcal{L}\|_{\max} + \|\nabla_{x_\tau} \mathcal{L}\|_{\max}) \tag{44}$$

$$\le (L \|w_\tau - x_\tau\|_F) \cdot (M + M) = 2LM \|w_\tau - x_\tau\|_F$$

Summing over $\tau$ with the bias-corrected weights gives $\|B(W) - B(X)\|_F \le 2LM \|W - X\|_F$. Substituting this into Eq. (43):

$$\left\| \frac{1}{\sqrt{B(W)+\epsilon}} - \frac{1}{\sqrt{B(X)+\epsilon}} \right\|_F \le \frac{2LM}{2\epsilon^{3/2}} \|W - X\|_F = \frac{LM}{\epsilon^{3/2}} \|W - X\|_F \tag{45}$$

**Final Result**. We now substitute the bounds from Eq. (39), Eq. (40), Eq. (41), and Eq. (45) into our main inequality Eq. (36).

$$\|g_{t-1}(W) - g_{t-1}(X)\|_F \leq \left(\frac{1}{\sqrt{\epsilon}}\right) \cdot (L\|W - X\|_F) + (M) \cdot \left(\frac{LM}{\epsilon^{3/2}}\|W - X\|_F\right)$$
$$= \left(\frac{L}{\sqrt{\epsilon}} + \frac{M^2 L}{\epsilon^{3/2}}\right)\|W_{t-1} - X_{t-1}\|_F \tag{46}$$

This final result provides an upper bound for the difference in the Adam update step that depends only on the problem constants $L, M, \epsilon$.

### I.8 Upper Bound for the Difference of $g_t$ in AdamW

The AdamW update function $g_t$ can be decomposed into the Adam update term and a decoupled weight decay term:

$$g_t(W_t) = g_t^{\text{Adam}}(W_t) + \lambda_t w_t \tag{47}$$

where $\lambda_t$ is the weight decay coefficient. We analyze the norm of its difference using the triangle inequality:

$$\|g_t(W) - g_t(X)\|_F = \|(g_t^{\text{Adam}}(W) - g_t^{\text{Adam}}(X)) + \lambda_t(w_t - x_t)\|_F \tag{48}$$
$$\leq \|g_t^{\text{Adam}}(W) - g_t^{\text{Adam}}(X)\|_F + \lambda_t\|w_t - x_t\|_F \tag{49}$$

We now substitute the final bound derived for the Adam component in Appendix I.7:

$$\|g_t^{\text{Adam}}(W) - g_t^{\text{Adam}}(X)\|_F \leq \left(\frac{L}{\sqrt{\epsilon}} + \frac{M^2 L}{\epsilon^{3/2}}\right)\|W_t - X_t\|_F \tag{50}$$

Assuming an upper bound for the weight decay coefficient, $\lambda_t \leq \lambda_{\max}$, and noting that $\|w_t - x_t\|_F \leq \|W_t - X_t\|_F$, we have:

$$\|g_t(W) - g_t(X)\|_F \leq \left(\frac{L}{\sqrt{\epsilon}} + \frac{M^2 L}{\epsilon^{3/2}}\right)\|W_t - X_t\|_F + \lambda_{\max}\|W_t - X_t\|_F \tag{51}$$
$$= \left(\lambda_{\max} + \frac{L}{\sqrt{\epsilon}} + \frac{M^2 L}{\epsilon^{3/2}}\right)\|W_t - X_t\|_F \tag{52}$$

This provides a rigorous upper bound for the difference in the AdamW update step.

