# OpenReview forum: "PASO: Step Parallel Stochastic Optimization"
_ICLR.cc/2026/Conference — Submitted to ICLR 2026_

### Official Review · Reviewer_VTKr · 2025-10-22

**Soundness:** 1
**Presentation:** 1
**Contribution:** 2
**Rating:** 2
**Confidence:** 4

**Summary:**

This paper reformulates sequential optimizers such as SGD and Adam as solving Triangular Nonlinear Equations (TNE) and proposes a general framework, PASO, that enables step parallelism.
Previous parallelization methods could not compute multiple optimization steps in parallel while preserving the exact gradient descent (GD) trajectory. In contrast to OptEx, which relies on approximation, PASO
aims to find exact points on the GD path by directly solving a system of equations.
The authors show that (i) the solution of the TNE uniquely matches the trajectory of sequential GD (Prop. 1), (ii) fixed-point iteration (FPI) can converge to this trajectory in at most T steps (Prop. 2), and (iii) a sliding window of width p makes the approach compatible with practical resource limits.
Experiments on language modeling (GPT-2, Llama-3.2-1B on WikiText) and image classification (CIFAR-10, Tiny-ImageNet with CNN/ViT) demonstrate up to 91× fewer iterations and up to 7.5× wall-clock speed-up, without quality degradation.

**Strengths:**

Originality

The idea of breaking inter-step dependency through equation solving is novel. The authors correctly point out that existing parallelization methods have not been able to compute multiple steps in parallel while preserving the exact gradient descent (GD) trajectory. Their reformulation of optimization as Triangular Nonlinear Equations (TNE) and the use of Fixed-Point Iteration (FPI) to recover the sequential GD path provide a clear and distinctive contribution. The paper also appropriately positions Optex as related work, clarifying that while Optex relies on approximation, PASO aims for exact solutions.

Significance (Potential Impact)

Inter-step parallelism represents an orthogonal axis to data, model, and pipeline parallelism. It has strong implications for scalability, especially in communication-bound training environments, as suggested by the results in Table 1.

Positive Details

The notation table in Appendix D is well-organized, making the proofs easy to follow.

**Weaknesses:**

1. Lack of comparison with approximate step-parallel methods such as Optex (insufficient validation of novelty)

The authors themselves refer to Optex (Shu et al., 2024) in the Related Work section (p.3), acknowledging it as an approach that uses kernel methods to approximate future gradients and achieve approximate step parallelism. However, the paper does not provide an empirical comparison with Optex in terms of accuracy, speed, computational cost, or communication overhead.
 Since PASO emphasizes exactness, a quantitative comparison on the same hardware and dataset is essential to show its practical advantage over a lighter, approximate method like Optex. In cases where approximation errors are tolerable, Optex could be more practical. As it stands, the paper’s theoretical claims are not sufficiently supported by empirical evidence regarding their real-world implications.


2. Reproducibility and experimental design issues

Incomplete wall-time comparison. When reporting wall-clock performance, full environment details are necessary (CUDA/NCCL/driver versions, PyTorch/Transformers/Tokenizers versions, GPU/CPU/network topology, mixed-precision and gradient scaling settings, and synchronization handling). Currently, only the use of W&B sweeps is mentioned (Sec. 6.1), and the code is not publicly available, making replication difficult. It is also unclear whether baseline tuning and soundness checks (multiple seeds, LR/β/weight-decay sensitivity, clipping, and regularization) were properly validated.


3. Evaluation metric issue (language modeling).

The x-axis should use the number of consumed tokens (num_tokens) rather than iterations, which is now the standard in modern large-scale training. Please include plots using num_tokens to allow fair comparison with DDP and other baselines under the same computational budget. Also clarify how num_tokens was defined and whether it was kept consistent across all experiments.



4. Ambiguity in metric definitions

The terms “testing accuracy”, “test accuracy”, “training accuracy”, and “train accuracy” are used inconsistently throughout the paper. Moreover, it is unclear whether “accuracy” in the language modeling experiments refers to token-level accuracy. The authors should include precise mathematical definitions of all evaluation metrics in the main text (e.g., Fig. 2, Table 2). While the Evaluation Metrics section mentions testing accuracy and testing perplexity as the standard metrics, Figure 2 labels training accuracy, creating confusion.

5. Unrealistically low perplexity values

The reported perplexity values of PPL = 1.4–1.6 on WikiText (Table 2) are unusually low compared with widely reported community benchmarks.
https://learn.arm.com/learning-paths/mobile-graphics-and-gaming/build-llama3-chat-android-app-using-executorch-and-xnnpack/3-understanding-llama-models/
https://huggingface.co/fedric95/Meta-Llama-3.1-8B-GGUF

https://huggingface.co/Graphcore/gpt2-wikitext-103
https://huggingface.co/neulab/gpt2-finetuned-wikitext103




The authors should specify the complete evaluation setup, including the exact script, tokenizer, detokenization process, context length, data split, and whether WikiText-2 or WikiText-103 was used. Clear references to comparable baselines under identical conditions are necessary. Currently, both the main text and Appendix only refer to the “WikiText dataset” in general terms, and even the footnote link does not clarify which variant was employed.


6. Clarity and consistency in figures and tables

Figure 2 (p. 7) appears to be a direct W&B dashboard capture; the lack of log scaling, consistent axes, or normalization makes it difficult to visually assess differences across methods. Furthermore, Tables 4–5 (p. 8) omit essential information such as model type, dataset, batch size, and learning rate in the captions, forcing the reader to refer back to the text to interpret results. Each table should explicitly note the model, dataset, batch size (B), and learning rate (η) for clarity.

7. Concerns about CV experiments

The reported accuracies on CIFAR-10/ViT are significantly below standard baselines
https://github.com/kentaroy47/vision-transformers-cifar10

Around 66% for CNN and 72% for ViT (Table 3, p. 9). It is also unclear whether these values represent top-1 or top-5 accuracy. Without proper baseline tuning and comparison to standard implementations, the validity of the proposed method’s improvements is difficult to assess.

8. Inconsistencies in model and implementation details

In Appendix B.1 (p. 16), footnote 1 links to the Chinese CLIP repository, making it unclear whether the experiments used a standard ViT, a CLIP-ViT variant, or a pretrained model. The correspondence between the text and the linked implementation should be clarified.

**Questions:**

1. Comparison with approximate step-parallel methods (e.g., Optex)

Please provide a fair comparison between PASO and Optex under identical hardware, dataset, and initialization conditions, evaluating accuracy (PPL/ACC), speed (steps and wall-time), and computational/communication cost (FLOPs, bytes).
It is important to clarify whether there exist realistic settings where the lighter, approximate Optex performs better when approximation errors are tolerable, and to what extent PASO’s theoretical exactness translates into practical performance advantages.
 The paper mentions that “both methods could be complementary (Optex can serve as an initializer),” but an experiment demonstrating this claim is required.

2. Ensuring reproducibility

 Please disclose full experimental details for wall-time measurements  including what was included or excluded (e.g., preprocessing, I/O, warm-up, synchronization)  and complete environment specifications (GPU/CPU types, network topology, CUDA/NCCL versions, PyTorch version, AMP configuration, and launch arguments). Public release of the wandb experiment logs would also be highly valuable.

3. Metric definitions and dataset specification

Clearly define the accuracy metric for language modeling (is it token-level?), and use consistent terminology across the paper (testing/test, training/train).
Specify which WikiText variant (2 or 103) was used, along with tokenizer, detokenization method, context length, and data split. Please include a table comparing PASO with prior baselines under identical conditions (related to Table 2).

4. Analysis using num_tokens as the x-axis

Define num_tokens, specify its values, and indicate whether it was kept consistent across all experiments.
Replot loss, PPL, and accuracy curves against num_tokens. Under an equal throughput constraint, please also compare increasing data parallelism versus increasing step parallelism (PASO) to illustrate trade-offs.

5. Soundness of CV and language modeling experiments

In Table 3, clearly specify whether the reported CIFAR-10 accuracy values are top-1 or top-5.
Re-tune the baselines (including learning-rate schedules, regularization, and data augmentation) so that they reach standard accuracy levels before comparing with PASO.
Same for language modeling tasks, please specify all the details and show existing comparable benchmark results or justify why the author achieved unrealistic lower perplexity.

6. Consistency of model references

In Appendix B.1, the ViT footnote links to the Chinese CLIP repository. Please clarify whether a plain ViT or CLIP-ViT was used, whether it was pretrained, and where the model weights originated.

---

> ### Author Response · Authors · 2025-11-24
>
> Thank you for your review.  **We have opened our code (see the general response)**.
>
> > *Lack of comparison with OptEx.  A lighter OptEx could be more practical.*
>
> **Empirical comparison**: We integrate our PASO into the official OptEx codebase. On OptEx's benchmark, PASO has a higher speedup than OptEx.
>
> Table 1:  Comparisons on OptEx benchmarks  for Ackley function (full details in App. B and `optex_cmp.py`)
> |Method|Optimality gap $\downarrow$|Iters $\downarrow$| Speedup $\uparrow$|
> |-|-|-|-|
> |Adam|$0.40\pm 0.18$|1000|$1.0\times$|
> |OptEx|$0.40\pm 0.06$|162|$6.17\times$|
> |PASO|$0.40\pm 0.11$|86|$11.63\times$|
>
> **Theoretical comparison**: A precise cost comparison is challenging due to vastly different engineering implementations. We thus conducted a more fundamental complexity analysis (full details in App. A).
>
> Table 2: Total per-iteration complexity comparison. $N$ GPU count, $d$ model dimension, $t_{cm}$ the comm. time, $T_0$, the number of storage historical gradients of OptEx.
> |Metric|OptEx|PASO|
> |-|-|-|
> |Comp. cost|$\mathcal{O}(T_0^3 + N T_0 (d + T_0))$|$\mathcal{O}(N^2 d)$|
> |Space cost|$\mathcal{O}(T_0 d + T_0^2 + N d)$|$\mathcal{O}(N d)$|
> |Comm. cost|$\mathcal{O}(t_{cm})$|$\mathcal{O}(t_{cm})$|
> |Speedup rate|$\mathcal{O}(\sqrt{N})$|$\mathcal{O}(N)$|
>
> This analysis reveals a critical practical bottleneck for OptEx that refutes the "lighter" assumption.
> - OptEx's space complexity scales with its $T_0$, making it **spatially infeasible for large-scale models** (where $d$ is massive). In contrast, PASO's is only $\mathcal{O}(Nd)$, which is the same as existing data parallelism.
> - PASO achieves a **linear speedup** rate of $\mathcal{O}(N)$ while OptEx only achieves a **sub-linear** speedup of $\mathcal{O}(\sqrt{N})$.
>
> These evidences strongly supports that PASO is  *a lighter method with significantly higher efficiency and practical applicability than OptEx. **We highlight that PASO is the first step-parallel training method based on equation system solving. This inherently distinguishes it from OptEx and is a fundamental breakthrough in parallel training.**
>
> > *Reproducibility and soundness check issue.*
>
> Our public code provides full details like execution scripts, etc. (see `README.md`). We include rigorous validation on learning rate & batch size sensitivity analysis (Tab.5&Fig. 3 in paper)
>
> > *Include plots using num_tokens to allow fair comparison with DDP and other baselines under the same computational budget. Also clarify how num_tokens was defined and whether it was kept consistent across all experiments.*
>
> We define token count as tokens processed by the model per iteration, reporting the cumulative consumed tokens during training. Our naive PASO implementation is compared against both naive and fully optimized Data Parallelism (DP). **The results are really exciting**. Our naive PASO  consumes the minimum number of tokens but achieves a 2.3× speedup over the naive DP, matching industrial DP performance.
>
> Table 3: Comparisons on GPT-2 over WiKiText-2 (see `llm_v2.py`). Batch size 112; learning rate 6e-5
> |Method|PPL$\downarrow$|Total tokens$\downarrow$|Iters$\downarrow$|time (s)$\downarrow$|
> |-|-|-|-|-|
> |Adam|20.4|7348175|1k|694|
> |Adam+Naive DP|20.5|7348942|1k|440 (1.8$\times$)|
> |Adam+Full optimized industrial DP (Pytorch's `DDP`)|20.4| 7348942|1k|**133** (5.2$\times$)|
> |Adam+Naive PASO|20.5|**7322406**|**0.16k**|136 (5.1$\times$)|
>
> We have added loss curves w.r.t consumed tokens in Fig. 3 in the revised version.
>
> > *Ambiguity in metric definitions. Is it token-level accuracy?*
>
>  We'll keep them consistent. For LLM, it is token-level accuracy (see `llm_v2.py`).
>
> > *The low accuracy/perplexity.  Clarify setup; WikiText-2 vs.103 usage; comparisons under identical conditions; top-1 or top-5 accuracy?*
>
> The low accuracy on CV tasks is likely because our models were trained with a fixed learning rate (lr) (see `cv_v1.py`). After using a decaying lr (see `cv_v2.py`), our accuracy matches the existing benchmark.
>
> Table 4: Top 1 Accuracy (ACC) comparison for CNN on CIFAR10 (`cv_v2.py`).
> |Method|ACC $\uparrow$|Iters $\downarrow$|Speedup $\uparrow$|
> |-|-|-|-|
> |Adam|81.5 $\pm$ 0.22|60k|1 $\times$|
> |PASO|81.3 $\pm$ 0.13|9.8k| 6.1 $\times$|
>
> The low PPL results from the incorrect use of repetitive padding tokens for PPL computation (see `llm_v1.py`).  After fixing this bug (see `llm_v2.py`), our PPL shows a comparable value v.s. community benchmarks.
>
> Table 5: Comparisons for Llama-3.2-1B on WikiText-2 (`llm_v2.py`)
> |Method|PPL $\uparrow$|Iters $\downarrow$|Speedup $\uparrow$|
> |-|-|-|-|
> |Adam|12.5 $\pm$ 0.36|1000| 1 $\times$|
> |PASO|12.3 $\pm$ 0.40 |81| 12.3 $\times$|
>
> > *Direct W&B Fig.; difficult to assess differences across methods. Tab. 4–5 omit information.*
>
> We plot Fig. 2 ourselves (see `plot.py`).  We believe different methods are comparable as we compare them on the same metric. We've fixed Tab. 4–5.
>
> >  *ViT details.*
>
> A plain ViT model is trained *from scratch* (see `cv_v2.py`).

---

> ### Author Response · Authors · 2025-11-27
> **Looking forward to your further feedback**
>
> Dear Reviewer **VTKr**
>
> Thank you for taking the time to review our manuscript and for your valuable feedback. We have carefully addressed all the comments and concerns raised, as reflected in our detailed responses and the revised manuscript and supplementary material.
>
> We are looking forward to your further feedback.
>
> Best regards,
>
> The Authors

---

> > ### Author Response · Authors · 2025-11-28
> >
> > Dear Reviewer **VTKr**,
> >
> > We are looking forward to your further feedback.
> >
> > Best regards,
> >
> > The authors.

---

### Official Review · Reviewer_kux2 · 2025-10-31

**Soundness:** 2
**Presentation:** 3
**Contribution:** 2
**Rating:** 2
**Confidence:** 3

**Summary:**

The paper proposes PASO, a step-parallel training scheme that re-casts an optimizer’s autoregressive trajectory $\\{w_t\\}$ into a triangular system of nonlinear equations (TNEs). A fixed-point iteration (FPI) over this system enables computing gradients for many future steps in parallel (within a sliding window of size $p$), then update all corresponding weights simultaneously. The authors claim (i) the TNE has a unique solution matching the GD trajectory (Proposition 1), (ii) their FPI converges to the GD trajectory in at most $T$ iterations (Proposition 2), and (iii) large reductions in steps (up to 91$\times$) and moderate wall-clock speedups (to 7.5$\times$) on GPT-2, Llama-3.2-1B and small CV models.

**Strengths:**

- The paper is easy to follow. The TNE formulation and windowed FPI are described clearly, with an end-to-end algorithm and pipeline figure.

- Reported step reductions (12.6-19.2$\times$ on GPT-2, 12.6-14.5$\times$ on Llama-1B, up to 31$\times$ on CIFAR-10) and best-case wall-clock speedup of 7.5$\times$ are encouraging.

**Weaknesses:**

- Proposition 2 only shows one can reproduce GD in $\le T$ outer iterations, and there is no theorem that $K<T$ under identifiable conditions. This weakens the paper’s core claim of principled step-parallel speedup.

- The method either launches $T$ parallel forward/backward passes or uses a window of size $p$, still requiring $p$ simultaneous graphs, data loaders, and optimizer updates. The paper nevertheless claims "1 model + 1 optimizer" storage per device in Table 1, which appears optimistic for step-parallel execution that must stage $p$ batches and intermediate states.

- The reported accuracy and perplexity results are presented without multiple seeds or error bars. Furthermore, the paper does not include comparisons with strong system baselines such as fully optimized FSDP/ZeRO or tuned pipeline + DP implementations under comparable hardware and batch-throughput settings.

- The EMA-based $\delta$ rule has no accuracy guarantee, it is tuned by Wandb sweeps rather than relying on derived principle.

**Questions:**

- Under your best run, what are the measured peak GPU memory, activation footprint, and optimizer-state duplication for PASO vs. strong DP/PP/MP baselines? How are $p$ batches staged without extra model/optimizer replicas?

- The method samples $T$ future mini-batches $\\{\zeta_t\\}$ and treats them as fixed to enable parallelism, will that implicitly change the sampling process and may require staging many batches concurrently?

- How are the $T$ future mini-batches produced and pinned to devices? Is the distribution i.i.d. with replacement? Does pre-sampling introduce measurable drift?

- Could you report multiple seeds with error bars, and include the compute budget (FLOPs, tokens)?

---

> ### Author Response · Authors · 2025-11-24
>
> Thank you for the feedback. **We believe there are **key misunderstandings** about our method's memory and theoretical claims.**
>
> > *Prop.2 only shows  $K \le T$ iterations, no theorem shows that $K < T$*
>
> We clarify that Prop. 2 guarantees that PASO converges in $K \le T$ iterations, including the case $K < T$. However, to precisely guarantee $K <  T$ universally is theoretically infeasible. For general non-convex optimization, deriving such a bound is mathematically intractable and remains an open problem in the field. Even for standard optimization methods, convergence rates are typically derived under strong assumptions (e.g., convexity) that do not hold for ML.
>
>
> > *The method requires $p$ simultaneous graphs. The paper nevertheless claims "1 model + 1 optimizer" storage per device in Tab. 1, which appears optimistic. Will that implicitly change the sampling process and may require staging many batches concurrently? How are the future mini-batches produced and pinned to devices? Is the distribution i.i.d. with replacement? Does pre-sampling introduce measurable drift?*
>
> We clarify that our "1 model + 1 optimizer" claim is **accurate and realized in our implementation**. As shown in `cv_nccl.py`, each rank instantiates exactly one model (`model = CNN().to(device)`) and one optimizer (`paso_optimizer`). At any iteration $k$, rank $r$ is assigned a specific time step $t+i$. It loads **only** the data batch corresponding to step $t+i$ and computes gradients for that specific step. Thus, the memory cost per device is strictly 1 model and 1 optimizer, identical to standard SGD and data parallelism.
>
> Regarding sampling, we implement a `PASODataLoader` that wraps standard PyTorch loaders to avoid the memory overhead of staging concurrent batches. Instead of pre-loading tensors, we pre-cache lightweight batch indices, allowing each device to retrieve the specific mini-batch required for its assigned time step $t+i$ on demand. **This guarantees that the data distribution remains identical to standard sequential training (i.i.d.) and introduces no statistical drift or additional memory footprint.** For instance, consider a  `PASODataLoader`  containing 3 batches, $\zeta_{0}, \zeta_{1}, \zeta_{2}$, across 6 sequential steps ($t=0$ to $t=5$).
>
> - **Epoch 1** Steps $t=0, 1, 2 $ correspond to $\zeta_{0}, \zeta_{1}, \zeta_{2}$.
> - **Epoch 2** Steps $t=3, 4, 5$ correspond to  $\zeta_{0}, \zeta_{1}, \zeta_{2}$.
>
> When a worker computes the gradient at step $t=0$, it retrieves $\zeta_{0}$ from its local dataloader. When computing the gradient at $t=4$, it retrieves $\zeta_{1}$ since $4 \pmod 3 = 1$.
>
> > *What is the  GPU memory for PASO vs. strong DP/PP/MP baselines? Furthermore, the paper does not include comparisons with strong system baselines such as fully optimized ZeRO...*
>
> We clarify that directly comparing our naive PASO implementation that doesn't incorporate any optimizations on the memory with fully optimized industrial parallel methods  **would be highly unfair and unreasonable**.   Our PASO has the same memory cost as naive data parallelism (DP), storing a single model/optimizer per GPU (see`cv_nccl.py`).
>
> To this end, we compare our naive PASO with naive DP  and its fully optimized implementation.
>
> **The results are really exciting**. It confirms our naive implementation consumes the minimum number of tokens but achieves a 2.3× speedup over naive DP, matching industrial DP performance. We believe that by applying equivalent industrial-grade optimizations, PASO can outperform the industrial methods. Crucially, PASO's inter-step parallelism complements existing intra-step parallel methods. Their synergy can yield more acceleration.
>
> Table 2: Comparisons on GPT-2 over WiKiText-2 (see `llm_v2.py`).
> |Method|PPL$\downarrow$|Total tokens$\downarrow$|Iters$\downarrow$|time (s)$\downarrow$|
> |-|-|-|-|-|
> |Adam|20.4|7348175|1k|694|
> |Adam+Naive DP|20.5|7348942|1k|440 (1.8$\times$)|
> |Adam+Full optimized industrial DP (Pytorch's `DDP`)|20.4| 7348942|1k|**133** (5.2$\times$)|
> |Adam+Naive PASO|20.5|**7322406**|**0.16k**|136 (5.1$\times$)|
>
> > *The  accuracy and perplexity w/o error bars.*
>
> In the revised version, we run all experiments with 5 different random seeds and update Tables 2&3 to report the mean and standard deviation.
>
> > *The EMA-based tolerance update rule $\delta$ has no accuracy guarantee; it is tuned by Wandb sweeps rather than relying on a derived principle.*
>
> We clarify that using EMA for adaptive estimation is a well-established paradigm in deep learning (e.g., the moving average of gradients in Adam). These methods prioritize adaptive smoothing over static theoretical bounds, which often fail to capture the dynamic variance of real-world training. Besides, deriving an exact principle for $\delta$ would require assumptions about the loss landscape (e.g., Lipschitz continuity constants) that are unknown or fluctuate during training. Thus, an empirical EMA adaptation is more practical and effective.

---

> ### Author Response · Authors · 2025-11-26
>
> Dear Reviewer **kux2**,
>
> We sincerely appreciate the time and effort you have devoted to reviewing our manuscript.
>
> We now have more exciting experimental results on the peak memory.
>
> Specifically, the table below compares the peak memory consumption of PASO against sequential baselines when increasing the window size $p$. Notably, PASO demonstrates comparable—and frequently slightly lower—peak memory usage relative to standard Adam and SGD. This memory efficiency stems from our implementation design (see `cv_v2.py`), where parallel worker processes are stateless: they are strictly responsible for computing gradients at specific time steps and do not maintain optimizer states (e.g., momentum or variance buffers) or historical artifacts. Consequently, the memory footprint does not scale linearly with the window size $p$, allowing PASO to leverage parallelism without incurring significant memory overhead.
>
> **Table 1: The peak memory (G) of the ViT model. $B=2048, \eta=1e-5$.**
>
> | Method | p=8 | p=40 | p=80 | p=120 |
> | :--- | :---: | :---: | :---: | :---: |
> | **Adam** | 27.9521 | 27.9521 | 27.9521 | 27.9521 |
> | **PASO** | 27.3330 | 27.5166 | 27.7061 | 27.8291 |
> | **SGD** | 27.9385 | 27.9385 | 27.9385 | 27.9385 |
> | **PASO** | 27.3310 | 27.5147 | 27.7030 | 27.8525 |
>
> For now, we have both theoretical and experimental results that confirm the per-device memory of our method is equivalent to standard sequential training. Consequently, we have made sure to address your remaining concerns directly and thoroughly.
>
> We understand that you may be handling multiple papers and have a busy schedule.
> However, we eagerly await your feedback on our responses.
>
> Best regards,
>
> The Authors

---

> ### Author Response · Authors · 2025-11-27
> **Looking forward to your further feedback**
>
> Dear Reviewer **kux2**
>
> Thank you for taking the time to review our manuscript and for your valuable feedback. We have carefully addressed all the comments and concerns raised, as reflected in our detailed responses and the revised manuscript and supplementary material.
>
> We are looking forward to your further feedback.
>
> Best regards,
>
> The Authors

---

> > ### Comment · Reviewer_kux2 · 2025-11-27
> >
> > The author's rebuttal addressed many of my concerns, and I have no further questions. I will raise the rating to 4.

---

> > > ### Author Response · Authors · 2025-11-28
> > > **Thanks for your further feedback**
> > >
> > > Dear Reviewer **kux2**
> > >
> > > We are pleased to learn that most of your concerns have been addressed and that you have raised the rating to 4. However, as a rating of 4 is still considered negative, we would greatly appreciate it if you could elaborate further on the reasons behind this evaluation. Our goal is to fully address all of your concerns, and your clarification will help us improve the manuscript accordingly.
> > >
> > > Thank you for your time.
> > >
> > > The Authors

---

> > > ### Author Response · Authors · 2025-11-28
> > >
> > > > *The author's rebuttal addressed many of my concerns, and I have no further questions. I will raise the rating to 4.*
> > >
> > > Dear Reviewer kux2,
> > >
> > > Thank you for your response and for acknowledging that our rebuttal has successfully addressed your concerns. We truly appreciate you engaging with our new experiments.
> > >
> > > However, we notice that while your latest comment states that your prior concerns have been resolved and you have no remaining questions, the rating remains at a 4 (Reject). A score of rejection typically indicates specific, identifiable weaknesses (e.g., lack of novelty), yet your text suggests these weaknesses have been rectified.
> > >
> > > We treated your initial feedback with the utmost seriousness. Over the past two weeks, we dedicated significant efforts to implementing new baselines from scratch and conducting extensive experiments to satisfy your requests. We did this based on the understanding that providing this evidence would lead to an objective re-evaluation of the paper’s merit.
> > >
> > > **To have these concerns resolved yet receive a rejection rating without a stated justification is disheartening**. As noted, PASO proposes **the first-ever step-parallel training scheme, overcoming the fundamental limitation of sequential gradient descent**. Given that the technical hurdles you identified are now cleared, we respectfully ask that you reconsider whether a score of 4 accurately reflects the current state of the manuscript.
> > >
> > > Best regards,
> > >
> > > The Authors

---

### Official Review · Reviewer_ypoz · 2025-11-01

**Soundness:** 3
**Presentation:** 3
**Contribution:** 3
**Rating:** 6
**Confidence:** 3

**Summary:**

This paper proposes PASO (Step Parallel Stochastic Optimization), a new optimization framework that aims to break the sequential dependency of gradient descent (GD).
The key idea is to reformulate the autoregressive GD process as solving a system of Triangular Nonlinear Equations (TNEs), which enables parallel gradient computation across multiple steps of optimization.
The authors provide theoretical guarantees showing that the TNE system admits a unique solution equivalent to the GD trajectory and that convergence can be achieved in equal or fewer iterations.
Empirically, PASO reportedly reduces the number of gradient descent steps by up to 91× and the wall-clock training time by up to 7.5×, without degrading model performance, on tasks such as image classification (CIFAR-10, Tiny-ImageNet) and language modeling (GPT-2, Llama-3.2-1B).

**Strengths:**

1. **Strong Conceptual Novelty**
   The paper tackles a long-standing challenge — the inherently sequential nature of gradient descent — from a completely different perspective.
   The formulation of GD as a **triangular nonlinear system** is conceptually elegant and opens a new avenue for step-level parallelization.

2. **Solid Theoretical Foundations**

   * The authors rigorously prove **the uniqueness** of the TNE solution (Proposition 1) and **the convergence** of the fixed-point iteration (Proposition 2).
   * The theoretical framework is mathematically consistent and supported by detailed appendices.

3. **Practical Implementation and Evaluation**

   * PASO is evaluated on both **vision** and **language modeling** tasks, including **LLMs up to 1B parameters**, demonstrating scalability.
   * The experiments show large reductions in iterations and training time while maintaining comparable accuracy, perplexity, and F1-scores.

4. **Compatibility and Orthogonality**
   PASO is **orthogonal to existing parallelism methods** (data, model, and pipeline parallelism) and can be combined with any optimizer (SGD, Adam, AdamW).

5. **Clear Reproducibility Commitment**
   The paper includes an explicit reproducibility statement, open-source promise, and detailed hyperparameter settings — all aligning with ICLR’s reproducibility expectations.

**Weaknesses:**

1. **Empirical Validation Scope**
   While experiments on GPT-2 and Llama-3.2-1B are impressive, the study lacks **large-scale validation** beyond a few models and datasets.
   There is no ablation on distributed setups beyond 8 GPUs, which limits the claim of scalability.

2. **Hardware and Communication Bottlenecks**
   The paper acknowledges major runtime limitations due to inefficient inter-GPU communication and CPU-mediated data transfer.
   Without addressing these, the reported speedups might not generalize to industrial-scale systems.

3. **Clarity of Algorithmic Description**
   Algorithm 1 and related sections are dense and could benefit from clearer pseudocode or diagrams to make the update mechanism and window-sliding strategy easier to follow.

4. **Notation Complexity**
   The paper introduces many symbols (e.g., (ŵ_t^{(k)}), (η_t), (ζ_t), (F_t)) that are hard to track. A consolidated notation table is referenced but should be better integrated in the main text.

5. **Lack of Theoretical Comparison**
   Theoretical complexity comparison against classical parallel SGD variants (e.g., Hogwild!, DC-ASGD) is limited; more discussion on **communication complexity** and **gradient staleness** would strengthen the argument.

**Questions:**

1. How does PASO behave under **non-smooth losses** or **non-convex constraints** (e.g., in diffusion or reinforcement learning models)?
2. Can PASO be adapted to work with **second-order or implicit optimizers** (e.g., Shampoo, K-FAC)?
3. In practice, how does PASO handle **gradient noise accumulation** when the window size (p) grows large?
4. Are the theoretical guarantees affected when using mixed precision or quantized gradients?

---

> ### Author Response · Authors · 2025-11-24
>
> Thank you for the very positive and encouraging review.
>
> > *Empirical Validation Scope ...*
>
> We agree that scaling beyond 8 GPUs represents a critical next step. Given current computational constraints (limited to 8 GPUs), large-scale validation is reserved for future work. **Significantly, PASO introduces the first step-parallel training method without compromising model accuracy, a fundamental advancement in parallel training.**
>
> >  *Might not generalize to industrial-scale systems.*
>
> We clarify that this limitation stems from the engineering constraints of our research prototype, not any fundamental constraint of PASO. Our `cv_nccl.py` resolves this bottleneck in which each GPU   maintains only a single model and optimizer.
>
> >  *Clearer pseudocode; Notation Complexity.*
>
> We clarify that we have included pseudocode (Alg. 1) and a diagram (Fig. 1). Due to space limitations, the notations are shown in App. D.
>
> > *Theoretical comparison against classical parallel SGD variants ; more discussion on communication complexity and gradient staleness.*
>
> We clarify that our PASO method fundamentally differs from asynchronous parallel approaches. These methods achieve data parallelism by tolerating **stale gradients**, which can harm or alter the convergence path.  PASO operates as a **step-parallel** framework that inherently avoids gradient staleness (Prop. 1 & 2). Thus, asynchronous optimization techniques are not directly comparable to our approach. The most relevant comparison is OptEx, which similarly employs step-parallelism. We provide a theoretical comparison below.
>
> Table 2: Total per-iteration complexity comparison. Denote by $N$ GPU count, $d$ model dimension, $t_{comm}$ the comm. time, $T_0$ the storage historical gradients number of OptEx.
> |Metric|OptEx|PASO|
> |-|-|-|
> |Comp. cost|$\mathcal{O}(T_0^3 + N T_0 (d + T_0))$|$\mathcal{O}(N^2 d)$|
> |Space cost|$\mathcal{O}(T_0 d + T_0^2 + N d)$|$\mathcal{O}(N d)$|
> |Comm. cost|$\mathcal{O}(t_{comm})$|$\mathcal{O}(t_{comm})$|
> |Speedup rate|$\mathcal{O}(\sqrt{N})$|$\mathcal{O}(N)$|
>
> This analysis reveals a critical practical bottleneck for OptEx.
>
> - **Space Bottleneck**: OptEx's space complexity is $\mathcal{O}((T_0 + N)d)$, scaling with its required history gradient size $T_0$. In contrast, PASO's is only $\mathcal{O}(Nd)$.
> - **Practical Implication**: The $\mathcal{O}(T_0 d)$ term, which arises from storing $T_0$ historical gradients, makes OptEx **spatially infeasible for large-scale models** (where $d$ is massive), which is the primary advantage of our PASO.
> - **Faster speedup**: PASO achieves a **linear speedup** rate of $\mathcal{O}(N)$ while OptEx only achieves a **sub-linear** speedup of $\mathcal{O}(\sqrt{N})$.
>
> > *How does PASO behave under non-smooth losses or non-convex constraints?*
>
> We clarify that PASO specifically accelerates gradient-descent-based methods, which are inherently ill-suited for non-smooth objectives. Thus, non-smooth loss functions lie outside the scope of PASO.
>
> Regarding non-convex constraints, our experiments demonstrate PASO's compatibility, having validated it on the highly non-convex model. Extending the PASO to diffusion models or reinforcement learning is a promising research direction, which we defer to future work.
>
> > *Can PASO be adapted to work with second-order or implicit optimizers?*
>
> Yes. PASO is a general framework. As long as the optimizer's update rule, $g_\tau$, can be formulated， which we demonstrated for SGD, Adam, and AdamW, PASO can be applied. Adapting it to second-order methods would simply require correctly defining the more complex $g_\tau$ term.
>
> > *How does PASO handle gradient noise accumulation when the window size (p) grows large?*
>
> PASO does not suffer from noise *accumulation* in the traditional sense.  PASO converges to the exact trajectory defined by a fixed set of $p$ mini-batches. The "noise" from this specific batch set is fully incorporated. Once it converges, the window slides forward by $s$ steps, and *new* mini-batches are sampled for the new end of the window. This rapid re-sampling prevents the trajectory from diverging based on a single "unlucky" set of $p$ batches.
>
> > *Are the guarantees affected when using mixed precision or quantized gradients?*
>
> PASO is fully compatible with them. PASO's guarantees are about finding the unique solution to the equation system. If the system is defined using mixed-precision or quantized gradients, PASO will simply find the exact trajectory corresponding to that mixed-precision/quantized optimization process.

---

> ### Author Response · Authors · 2025-11-27
> **Looking forward to your further feedback**
>
> Dear Reviewer **ypoz**
>
> Thank you for taking the time to review our manuscript and for your valuable comments and recognition. We have carefully addressed all the comments and concerns raised, as reflected in our detailed responses and the revised manuscript and supplementary material.
>
> We are looking forward to your further feedback.
>
> Best regards,
>
> The Authors

---

### Official Review · Reviewer_bekR · 2025-11-03

**Soundness:** 2
**Presentation:** 3
**Contribution:** 2
**Rating:** 4
**Confidence:** 3

**Summary:**

The study presents the approach to avoid the autoregressive nature of the most gradient-based optimizers through the search of the entire optimization process trajectory via solving a triangular nonlinear system. The theoretical guarantees of equivalence between the trajectory induced by the proposed approach and the trajectory from the baseline optimizers are derived. The work also suggests a computationally efficient approach to parallelize the solving of the target system and a stopping criterion to prevent redundant iterations. The core idea is to use fixed-point iteration and to consider blocks of sequential equations in parallel. The empirical evaluation demonstrates that the proposed approach is faster in terms ot iterations and runtime than classical SGD and Adam optimizers on CV and NLP tasks with sufficiently large models and datasets.

**Strengths:**

The main strength of the presented manuscript is the nice idea of revisiting the classical concept of gradient-based optimizer from a non-trivial perspective. The proposed reformulation can provide an alternative direction for accelerating optimizers. The derived numerical results confirm the potential of this direction and the tractability of the stated nonlinear triangular system of equations. The theoretical justification for the coincidence between the trajectories obtained from the autoregressive method and those from the proposed method explains the correctness of this reformulation. Last but not least, the effect of hyperparameters on the results presented in the main text is also investigated.

**Weaknesses:**

Although the presented approach demosntrates promising results, I have identified the following weaknesses:
1. The comparison of the **parallel** approach to solve the nonlinear triangular system is done only with the **sequential** basic optimizers. It is evident that the authors observe a significant speed-up. Comparing with standard frameworks for parallel training on multiple GPUs (e.g., DeepSpeed) would strengthen the contribution and highlight the effect of the non-autoregressive nature of the underlying process.
2. No memory overhead analysis is discussed in the study.
3. The convergence analysis of the Fixed-point iteration corresponding to $g$ from basic optimizers is not presented in the main text. Thus, it is unclear why the selected approach to solving the target system of equations is valid in such case. Proposition 2 claims the convergence but it could be arbitrary slow.
4. Algorithm 1 does not use $\delta$ for any stopping decision and does not update $t$ in inner iteration. In addition the role of $\delta$ is also missing in Figure 1, so the stopping criterion looks artificial and insufficiently mentioned in the context of the complete pipeline.

**Questions:**

1. What is $\xi_{\tau}$ in Eq. (3) ?
2. What is $r$ is Eq. (3) and how does it depend on batch size?
3. What is $\mathbf{p}$ in lines 308-309?
4. What is $\alpha$ in lines 306-307?
5. What is the reason of limitation the batch size to 30 for the LLaMa 3.2-1B model?
6. Do you have any ideas how to combine your approach with existing parallelism techniques? For example, how to use your approach for a such large model that storage gradient cprrespoinding to the $i$-th equation in the **single** GPU becomes infeasible?

---

> ### Author Response · Authors · 2025-11-24
>
> We thank Reviewer bekR for the review. However, **we found that the reviewer had overlooked many details that had been clearly discussed in the manuscript.**
>
> > *Comparing with parallel training on multiple GPUs.*
>
> We clarify that directly comparing our naive PASO implementation that doesn't incorporate any optimizations on the memory with fully optimized industrial parallel methods  **would be highly unfair and unreasonable**. To make a fair comparison, we compare our naive PASO with naive data parallelism (DP)  and its fully optimized implementation.
>
> **The results are really exciting**. It confirms our naive implementation of PASO  consumes the minimum number of tokens but achieves a 2.3× speedup over naive DP, matching industrial DP performance. **We believe that by applying equivalent industrial-grade optimizations, PASO can significantly outperform the industrial methods.** Crucially, PASO's inter-step parallelism complements existing intra-step parallel methods. Their complementary nature can yield more acceleration when combined.
>
> Table 1: Comparisons on GPT-2 over WikiText-2 (see `llm_v2.py`).
>
> |Method|PPL$\downarrow$|Total tokens$\downarrow$|Iters$\downarrow$|time (s)$\downarrow$|
> |-|-|-|-|-|
> |Adam|20.4|7348175|1k|694|
> |Adam+Naive DP|20.5|7348942|1k|440 (1.8$\times$)|
> |Adam+Full optimized industrial DP (Pytorch's `DDP`)|20.4| 7348942|1k|**133** (5.2$\times$)|
> |Adam+Naive PASO|20.5|**7322406**|**0.16k**|136 (5.1$\times$)|
>
> > *No memory overhead analysis is discussed in the study.*
>
> We clarify that Tab. 1 states the overhead is "1 model + 1 optimizer" per device.  We further give memory overhead using space complexity below (Full details in Appendix A).
>
> Table 2: Total per-iteration complexity comparison.  $N$ GPU count; $d$ model dimension; $t_{comm}$ the comm. time; $T_0$, the storage historical gradients number of OptEx.
> |Metric|OptEx| our PASO|
> |-|-|-|
> |Comp. cost|$\mathcal{O}(T_0^3 + N T_0 (d + T_0))$|$\mathcal{O}(N^2 d)$|
> |Space cost|$\mathcal{O}(T_0 d + T_0^2 + N d)$|$\mathcal{O}(N d)$|
> |Comm. cost|$\mathcal{O}(t_{comm})$|$\mathcal{O}(t_{comm})$|
> |Speedup rate|$\mathcal{O}(\sqrt{N})$|$\mathcal{O}(N)$|
>
> This analysis reveals a critical practical bottleneck for OptEx.
>
> - **Space Bottleneck**: OptEx's space complexity is $\mathcal{O}((T_0 + N)d)$, scaling with its required history gradient size $T_0$. In contrast, PASO's is only $\mathcal{O}(Nd)$.
> - **Practical Implication**: The $\mathcal{O}(T_0 d)$ term, which arises from storing $T_0$ historical gradients, makes OptEx **spatially infeasible for large-scale models** (where $d$ is massive), which are the primary advantage of our PASO.
> - **Faster speedup**: PASO achieves a **linear speedup** rate of $\mathcal{O}(N)$ while OptEx only achieves a **sub-linear** speedup of $\mathcal{O}(\sqrt{N})$.
>
> > *The convergence analysis corresponding to $g$  is not presented in the main text. Prop. 2 claims the convergence, but it could be arbitrarily slow.*
>
> We clarify that the main text in Prop. 2 gives the convergence analysis for $g$ from different optimizers. Prop. 2 doesn't show our method is arbitrarily slow. It only confirms our method never converges more slowly than sequential baselines. Our experiments demonstrate that PASO converges extremely rapidly.
>
> > *Alg. 1 and Fig. 1 don't use $\delta$. Doesn't update $t$ in inner iteration*
>
> The  $\delta$ is **explicitly used**.
>
> 1. **Alg.1**: $\delta$ is used by mentioning Eq. 9  (i.e., $d(\cdot) \le \delta$ in Eq.9).
> 2. **Fig. 1**: $\delta$ are shown in the box labeled "Check error and calculate stride" and "Adaptive tolerance update".
>
>  $t$ is **explicitly** updated on Line 11 of  Alg. 1.
>
> > *What is $r$, $g_\tau$, $N$, $\alpha$?*
>
> We clarify that we have **clearly defined** these notations
>
> - **$r$**  is the number of history weights used by the optimizer (see App. G).
> - **$g_\tau$** is the general gradient update term for a specific optimizer. It is defined for SGD in Eq. (4) and for Adam in Eq. (15).
> - **$N$** (Tab. 1) is the number of GPUs.
> - **$\alpha$** (Tab. 1) is the communication-to-computation time ratio.
>
> > *What is the reason of limiting the batch size to 30 for the LLaMa 3.2-1B model?*
>
> Our current implementation (see `llm_v2.py`) is memory-intensive, in which we adopt a master-worker multiprocessing way where the master maintains a window of the model. Therefore, for the 1B Llama model, the maximum batch size is limited to 30 on our GPU. However, we want to highlight that this is an engineering bottleneck, not a fundamental constraint of the PASO algorithm. Our  `cv_nccl.py` explores a memory-efficient method where only a single model and optimizer are maintained per GPU.
>
> > *Do you have any ideas how to combine your approach with existing parallelism techniques?*
>
> For a model too large for one GPU, one would *first* apply **intra-step** parallelism to shard the model *within* a single step. PASO would then operate *on top* of this, managing the **inter-step** parallelism.

---

> ### Author Response · Authors · 2025-11-27
> **Looking forward to your further feedback**
>
> Dear Reviewer **bekR**
>
> Thank you for taking the time to review our manuscript and for your valuable comments. We have carefully addressed all the comments and concerns raised, as reflected in our detailed responses and the revised manuscript and supplementary material.
>
> We are looking forward to your further feedback.
>
> Best regards,
>
> The Authors

---

### Author Response · Authors · 2025-11-24
**General Response to all Reviewers.**

# General Response to all Reviewers.

We sincerely thank the reviewers for their time and effort in carefully reviewing this manuscript. We are encouraged by the exceptionally positive assessment on the **strong conceptual novelty** (Reviewers bekR, ypoz, VTKr), **solid theoretical foundations** (Reviewers bekR, ypoz), and **promising speedup results** (Reviewers bekR, kux2).

The reviewers have raised several concerns regarding reproducibility, comparisons relative to existing parallel methods (such as OptEx and strong parallel methods), costs, and the low accuracy and perplexity. In response, **we have some exciting theoretical and experimental results** that have directly and thoughtfully addressed these concerns:

* **Reproducibility**: We have anonymously **open-sourced our code** with a detailed running script, full environment specifications (including CUDA/NCCL versions, hardware topology), library dependencies, and random seed configurations (Reviewers VTKr, kux2). **Our code: https://anonymous.4open.science/r/PASO-8ECE/.**
* **Computation, Communication, and Space Complexity:** We have added both theoretical complexity analyses and empirical comparisons against the approximate method, OptEx, in Appendix A. **Excitingly**, our analysis reveals that PASO achieves a linear speedup rate, whereas OptEx is limited to a sub-linear. Furthermore, PASO maintains significantly lower space complexity ($\mathcal{O}(Nd)$)  and compared to OptEx ($\mathcal{O}((T_0+N)d)$).  In our memory-efficient implementation `cv_nccl.py`, PASO maintains only one optimizer and model per device, matching the memory footprint of data parallelism. These make PASO fundamentally scalable for large models.
* **Competitive Performance against Fully Optimized Parallel Baselines:** We compared our naive PASO implementation against both naive Data Parallelism (DP) implementation and fully optimized industrial DP (PyTorch `DDP`). **Remarkably**, our naive PASO implementation is $2.3$ times faster than the naive DP, and matches the performance of a fully optimized industrial DP framework, Pytorch's `DistributedDataParallel`, achieving a $5.1\times$ speedup while consuming the minimum number of tokens. This highlights the immense potential of PASO.
* **Validation under Standard Implementation:** We have addressed the implementation issues regarding low accuracy and perplexity calculations. Our updated results, now reported with multiple random seeds and standard deviations, **match community benchmarks**. This confirms that PASO accelerates training without compromising model quality.

**We sincerely invite you to refer to the detailed responses below and the provided source code. We are genuinely excited about these findings,** as they validate **PASO as a fundamental advancement in parallel training**, introducing **the first step-parallel training method without the loss of quality** based on equation system solving. What makes PASO truly compelling is **its unique role as an inter-step parallel approach, which perfectly complements existing methods confined to intra-step parallelism.** This **orthogonality** creates a powerful synergy, unleashing new levels of parallel efficiency for current methods. For instance, one can seamlessly apply intra-step parallelism to shard the model within a single step, while PASO operates harmoniously on top, orchestrating the inter-step parallelism.

**Looking ahead**, our vision is to evolve PASO into a robust, industrial-caliber step-parallel training ecosystem. By synergizing our step-parallel approach with existing parallelism and memory optimization techniques like ZeRO, we aim to unlock unprecedented efficiency in the training of LLMs.

**Last but not least**, we thank the PCs, ACs, and reviewers again for their invaluable time and effort. We commit that the final accepted manuscript will include the newly added experiments and analyses. Furthermore, **we will ensure the reviews and author discussion remain public as well.**

---

### Author Response · Authors · 2025-11-25
**Looking forward to your further assessment**

Dear **Reviewers**,

Thank you for taking the time to review our manuscript and for your valuable feedback. We have carefully addressed all the comments and concerns raised, as reflected in our detailed responses and the revised manuscript and supplementary material.

We sincerely appreciate your efforts and look forward to your further assessment.

Best regards,

The Authors

---

### Meta-Review · Area_Chair_5Vvv · 2026-01-12

**Summary:**

Reviewers generally agree that this paper is novel in reframing step-level optimization as solving a triangular nonlinear system and the theoretical reformulation is interesting.
However, the concerns that the practical impact and empirical validation do not fully support the strength of the claims.
Three reviewers assigned rejection scores, and their concerns include:
1. experimental soundness: initially unrealistic LM perplexities and unusually low CV accuracies, ambiguous metric definitions, incomplete token-budget reporting
2. lack of comparisons against strong modern training systems
3. uncertainty about whether the method delivers principled and robust gains beyond prototype settings given communication bottlenecks and reliance on heuristic stopping.

The remaining reviewer assigned a score of borderline acceptance, but still highlighted missing comparisons and limited large-scale validation as key weaknesses.

**Reviewer Concerns:**

Addressed concerns:

- Reproducibility: authors released code and provided additional implementation details.

- Metric irregularities: authors acknowledged and fixed issues leading to unrealistic PPL and low CV accuracy and clarified metric definitions.

- Error bars and multiple seeds: authors reported details.

- Memory overhead skepticism: authors provided peak memory measurements.

Outstanding concerns:

- Strength of systems evidence vs. claims: The paper argues for a broadly impactful step-level parallel training scheme, but still lacks convincing evaluation against strong, realistic training stacks and lacks full accounting of what is included in wall-time. The “prototype against industrial baselines” argument does not fully resolve the concern.

- Credibility impact of post-hoc fixes: While the bug fixes are appreciated, the fact that headline numbers depended on evaluation/training issues reduces confidence in the overall experimental pipeline.

- Scaling scope: Limited distributed validation (not beyond 8 GPUs) remains a gap relative to scalability claims.

**Reviewer Scores:**

- Reviewer ypoz : Likely unchanged at 6. The concerns of empirical validation scope and communication bottlenecks stand.

- Reviewer bekR : May be unchanged at 4. The proposed scaling scope concern is not fully addressed.

- Reviewer kux2 : From 2 to 4 in discussion.

- Reviewer VTKr : Could be 4 or 2. I recognize the reviewer's concern about experimental soundness.

---

### Decision · Program_Chairs · 2026-01-26

Reject